

# Driving parameters of biogenic volatile organic compounds and consequences on new particle formation observed at an Eastern Mediterranean background site

Cécile Debevec[1], Stéphane Sauvage[1], Valérie Gros[2], Karine Sellegri[3], Jean Sciare[4,2], Michael Pikridas[4], Iasonas Stavroulas[4], Thierry Leonardis[1], Vincent Gaudion[1], Laurence Depelchin[1], Isabelle Fronval[1], Roland Sarda-Esteve[2], Dominique Baisnée[2], Bernard Bonsang[2], Chrysanthos Savvides[5], Mihalis Vrekoussis[4,6], Nadine Locoge[1].

[1] IMT Lille Douai, Univ. Lille, SAGE - Département Sciences de l'Atmosphère et Génie de l'Environnement, 59000 Lille, France
[2] Equipe CAE, Laboratoire des Sciences du Climat et de l'Environnement (LSCE), Unité Mixte CEA-CNRS-UVSQ, Gif sur Yvette, 91190, France
[3] Laboratoire de Météorologie Physique (LaMP), CNRS UMR 6016, Université Blaise Pascal, Aubière, 60026, France
[4] Energy, Environment and Water Research Centre, the Cyprus Institute (CyI), Nicosia, 2121, Cyprus
[5] Department of Labour Inspection (DLI), Ministry of Labour, Welfare and Social Insurance, Nicosia, 1493, Cyprus
[6] Institute of Environmental Physics (IUP), University of Bremen, Bremen, 28359, Germany

Correspondence to: Stéphane Sauvage (stephane.sauvage@imt-lille-douai.fr) – Cecile Debevec (cecile.debevec@imt-lille-douai.fr)

**Abstract.** As a part of the ChArMEx (Chemistry Aerosol Mediterranean Experiments) and ENVI-Med CyAr (Cyprus aerosols and gas precursors) programs, this study aims primarily at providing an improved understanding of the sources and the fate of volatile organic compounds (VOCs) in the Eastern Mediterranean. More than 60 VOCs, including biogenic species (isoprene and 8 monoterpenes) and oxygenated VOCs were measured during a 1-month intensive field campaign performed in March 2015 at the Cyprus Atmospheric Observatory (CAO), a regional background site in Cyprus. VOC measurements were conducted using complementary on-line and off-line techniques. Biogenic (B)VOCs were principally imputed to local sources and characterized by compound-specific daily cycles such as diurnal maximum for isoprene and nocturnal maximum for α,β-pinenes, in connection with the variability of emission sources. The simultaneous study of pinenes and isoprene temporal evolution and meteorological parameters has shown that BVOC emissions were mainly controlled by ambient temperature, precipitation and relative humidity. It was found that isoprene daytime emissions at CAO depended on temperature and solar radiation changes whereas nocturnal BVOC concentrations (e.g. from oak and pine forests) were more prone on the relative humidity and temperature changes. Significant changes in monoterpene mixing ratios occurred during and after rain. The second part of the study focused on new particle formation events (NPF) at CAO. BVOCs are known to potentially play a role in the growth as well as in the early stages of formation of new atmospheric particles. Based on observations of the particle size distribution performed with a differential mobility particle sizer (DMPS) and the total number concentrations of particles larger than 1 nm diameter measured by particle-size magnifier (PSM), NPF





events were found on 14 out of 20 days of the field campaign. For all possible proxy parameters (meteorological parameters, calculated $H_2SO_4$ and measured gaseous compounds) having a role in NPF, we present daily variations of different classes during nucleation events and non-event days. NPF can occur at various condensational sink (CS) values and both under polluted and clean atmospheric conditions. High $H_2SO_4$ concentrations coupled with high BVOC concentrations seemed to

be one of the most favorable conditions to observe NPF at CAO in March 2015. NPF event days were characterized by either (1) a predominant anthropogenic influence (high concentrations of anthropogenic source tracers observed), (2) a predominant biogenic influence (high BVOC concentrations coupled with low anthropogenic tracer concentrations), (3) a mixed influence (high BVOC concentrations coupled with high anthropogenic tracer concentrations) and (4) a marine influence (both low BVOC and anthropogenic concentrations). More pronounced NPF events were identified during mixed

anthropogenic-biogenic conditions compared to the pure anthropogenic or biogenic ones, for the same levels of precursors. Analysis of specific NPF periods of the mixed influence type highlighted that BVOC interactions with anthropogenic compounds enhanced nucleation formation and growth of newly formed particles. During these days, the nucleation mode particles may be formed by the combination of high $H_2SO_4$ and isoprene amounts, under favorable meteorological conditions (high temperature and solar radiation and low relative humidity) and low CS. During the daytime, growth of the newly

formed particles, sulfate but also oxygen-like organic aerosol (OOA) mass contributions increased in the particle phase. High BVOC concentrations were observed during the night following NPF events, accompanied with an increase of the CS and of semi volatile OOA contribution, suggesting further BVOC contribution to aerosol nighttime growth by condensing onto pre-existing aerosols.

# 1 Introduction

The Mediterranean atmosphere is strongly afflicted by particulate and gaseous pollutions at once. Consequently, aerosol and/or ozone mixing ratios are usually elevated in the Mediterranean than in the majority of the continental European regions, chiefly during summer (Doche et al., 2014; Menut et al., 2015; Nabat et al., 2013; Safieddine et al., 2014). The Mediterranean is also regarded as a notorious climate change "hot spot" and which is foreseen to undergo significant warming and drying in the 21$^{st}$ century (Giorgi, 2006; Kopf, 2010; Lelieveld et al., 2014). This may have strong implications

on natural and anthropogenic emissions and their fate in the atmosphere with unpredictable impacts. Indeed, air composition, concentration levels, and trends in the Mediterranean region still remain arduous to evaluate mainly due to limited in-situ observation datasets. Supplementary information on the air chemical composition, including the speciation and the reactivity of volatile organic compounds (VOCs) at representative regional background sites, will further enhance our actual comprehension of the intricacy of the Mediterranean atmosphere. Given this background, the ChArMEx (the chemistry-

aerosol Mediterranean experiment, http://charmex.lsce.ipsl.fr) (Dulac, 2014) international project of the multidisciplinary regional research program MISTRALS (Mediterranean integrated studies at regional and local Scales; http://mistrals-home.org) proposes developing and coordinating regional research actions for a scientific evaluation of the present and



future state of the atmospheric environment in the Mediterranean basin, and of its incidences on the regional climate, air quality, and marine biogeochemistry.

Within the framework of ChArMEx and ENVI-Med CyAr programs, an intensive field campaign was carried out during a 1-month period (March 2015) at the Cyprus atmospheric observatory (CAO, http://www.cyi.ac.cy/index.php/cao.html) to provide insights of the origins and fates of VOCs and aerosols in the Eastern Mediterranean, focusing on an extensive high time resolution in-situ measurements performed at a representative receptor site. An important database combining gaseous and particulate observations was collected, including over 60 VOCs determined by various on-line and off-line techniques. The resulting dataset has been presented by details in Debevec et al. (2017). In that work, a Positive Matrix Factorization (PMF) analysis along with a concentration field (CF) analysis have been performed on a database containing 20 VOCs in order to better identify and characterize co-variation factors of VOCs. This study has highlighted aged or local primary emissions together with secondary photochemical transformations taking place during the transport of air masses. As presented in the latter study, and due to the background regional pattern of the measurement site, concentration levels of anthropogenic species were low (e. g., average mixing ratio of 299 and 114 ppt for ethylene and benzene, respectively), whereas significant levels of primary biogenic compounds locally emitted were observed. Oxygenated (O)VOCs were found to largely dominate the VOC budget and they were mainly explained by biogenic sources (64 %) according to Debevec et al. (2017). Thus, due to their significant contribution to the VOC budget in this environment, it is essential to characterize the biogenic emissions and better evaluate their impact in the Eastern Mediterranean.

Isoprene, terpenes (monoterpenes, sesquiterpenes) and OVOCs (alcohols, carbonyl compounds and organic acids) are the most common biogenic (B)VOCs reported in the literature (e. g., Bouvier-Brown et al., 2009; Llusia et al., 2012; Seco et al., 2011). Isoprene and monoterpenes are of major importance due to their significant emission rates in the atmosphere (Guenther et al., 2006; Helmig et al., 2013; Peñuelas and Staudt, 2010). BVOC emissions can be initiated or altered by a large number of factors such as both biotic and abiotic stress (Laothawornkitkul et al., 2009; Loreto and Schnitzler, 2010; Niinemets et al., 2004; Possell and Loreto, 2013), controlling the emissions of BVOCs to the atmosphere. In the atmosphere, (B)VOCs undergo fast reactions with hydroxyl radicals (OH), nitrate radicals ($NO_3$) and ozone ($O_3$) and can generate a variety of oxidized products, such as carbonyls, organic acids and alcohols, thus playing a significant role in the oxidative capacity of the atmosphere (Fuentes et al., 2000; Gelencsér et al., 2007; Helmig et al., 2006; Kanakidou et al., 2005). Undergoing multigenerational oxidation processes, reactions of BVOCs in the atmosphere lead to rising functionalized products with sufficiently low volatility (Aumont et al., 2012; Jimenez et al., 2009; Kroll and Seinfeld, 2008) to be involved in the formation of secondary organic aerosols (SOA) (Fuzzi et al., 2006; Kanakidou et al., 2005).

New particle formation (NPF) is a process traducing the secondary formation of atmospheric particles (Dal Maso et al., 2005). Although NPF is a global phenomenon observed in many different environments (Kulmala et al., 2004; Kulmala and Kerminen, 2008), strong uncertainties on the processes governing NPF are still remaining. Until recently, it was considered that NPF could not occur without the involvement of sulfuric acid ($H_2SO_4$) in the nucleation step as well as for



the growth of newly formed particles (Kulmala et al., 2013; Sipilä et al., 2010). However, it is now recognized that typical daytime $H_2SO_4$ concentrations are too low for sulfuric acid and water alone to account for the NPF rates observed in the lower atmosphere (Boy et al., 2003; Kirkby et al., 2011). A ternary compound is required to stabilize $H_2SO_4$ clusters, such as ammonia ($NH_3$) and amines, although the latter are probably not sufficient for reaching the observed NPF rates (Almeida et

al., 2013; Kirkby et al., 2011; Kürten et al., 2016). In the area of the eastern Mediterranean, seasonal variation of nucleation frequency has been explained by Pikridas et al. (2012) in function of the availability of gas-phase $NH_3$ transferred to the particulate phase to neutralize the aerosol population. Additionally, it is well established that oxidation products of VOCs are important for particle growth (Riipinen et al., 2011; Sellegri et al., 2005) and it has been recently shown that VOCs probably play a major role in nucleation step, especially BVOCs and their oxidation products (Riccobono et al., 2014; Schobesberger

et al., 2013; Zhao et al., 2013). There have been mixed reports regarding the role of VOCs in NPF. Some studies have shown that high isoprene concentrations can inhibit biogenic NPF by scavenging OH radicals (Kanawade et al., 2011; Kiendler-Scharr et al., 2009). However, Zhang et al. (2004) reported that the interaction between VOCs and $H_2SO_4$ can promote efficient formation of organic aerosols. Moreover, chamber experiments highlighted ion-induced nucleation of pure biogenic particles (mostly α-pinene) which it is believed to dominate nucleation in pristine environments where the condensational

sink (CS) levels are low and scavenging of these compounds on pre-existing particles is limited (Kirkby et al., 2016).

Therefore, in order to understand the role of BVOCs in atmospheric chemistry, it is important to study their emission drivers, atmospheric abundance, and to characterize their atmospheric oxidation. This paper will address these objectives and is organized as follow: first, Sect. 2 is dedicated to the sampling site description together with the different on-line/off-line analytical techniques. In Sect. 3.1, we examine primary BVOC levels and their temporal variations. In Sect.

3.2, temporal variations of the main monoterpenes and isoprene are compared with meteorological parameters to determine the dominant factors controlling BVOC emissions. Then, OVOC levels and their biogenic origins are discussed in Sect. 3.3. Finally, we investigate NPF observed at CAO in Sect. 3.4 with a focus on the role of BVOCs.

## 2 Material and Methods

### 2.1 Sampling site

Cyprus is an island located on the Eastern part of the Mediterranean Sea, 110 km southerly from the Turkish coast, c.a. 250 km westerly from Lebanon and Syria and 780 km easterly from Crete (Greece). This island covers an area of 9250 km$^2$ and includes 648 km of coastline. The major agglomerations of the island are namely, Nicosia, Limassol, Larnaca, Paphos, Famagusta and Kyrenia (321,816; 176,600; 84,591; 61,986; 50,265 and 33,207 inhabitants, respectively, census 2011 – Fig. 1). Air masses circulating over Cyprus are restrained by two mountain complexes, the Troodos and the Kyrenia

Mountains (located in the center and the north parts of Cyprus, respectively).

As a part of two French research programs, the ChArMEx and ENVI-Med CyAr (Cyprus aerosols and gas precursors), an intensive field campaign has been conducted at a regional background site of Cyprus (CAO, 33.05° E -




35.03° N, 532 m above sea level, a.s.l. - Sciare, 2016) from 1 March to 29 March 2015. CAO is a regional background station from the global atmospheric watch (GAW), and is operating under ACTRIS, the European research infrastructure for the observation of aerosol, clouds and trace gases (http://actris2.nilu.no/). The station is co-operated by the department of labour inspection (DLI) within the network of the "co-operative programme for monitoring and evaluation of the long-range

transmission of air pollutants in Europe" (EMEP). Consequently, criteria established by the EMEP, GAW and ACTRIS networks insure a high quality assurance for the atmospheric measurements performed at CAO. The station is located in the central area of the island about 20 km from the western coast and more than 35 km off the main Cypriot agglomerations, with limited influences of anthropogenic emissions from these cities. CAO is situated at the top of a hill (premises of the Cyprus Department of Forests) with no major local pollution sources (few car circulations during week days). The

measurement site is encompassed by widespread vegetation such as "maquis", shrubland characteristic of Mediterranean areas, and close to oak and pine forests covering the Troodos Mountain range (Fall, 2012), that are known as high emitters of BVOCs (Owen et al., 2001).

## 2.2 Experimental Set-up

### 2.2.1 VOCs measurements

Non-methane hydrocarbons (NMHCs) and OVOCs were measured employing complemental on-line and off-line techniques described in the following. The inlets were about 3 m above ground level (a.g.l.). Table 1 resumes the characteristics of the methods carried out during the campaign and indicates a list of the monitored VOCs.

**On-line VOC measurements:**

At a time resolution of 30 min, 20 VOCs, including $C_2$ to $C_{10}$ anthropogenic VOCs and $C_{10}$ BVOCs, were measured

using two automated gas chromatographs (GCs, Chromatotec, Saint-Antoine, France) outfitted with a flame ionization detector (FID). A detailed description of both instruments (ChromaTrap and AirmoVOC), sampling set up, technical information (pre-concentration, desorption-heating times, type of traps, column types) and the calibration procedure were given in Debevec et al. (2017). Very satisfactory detection limits (as 3 σ of the baseline) were found with values below 104 and 17 ppt for ChromaTrap and for AirmoVOC, respectively. Relative uncertainties of VOCs measured with ChromaTrap

analyzer typically ranged from 14 % (ethane) and 73 % (propene) and from 18 % (benzene) and 53 % (o-xylene) for VOCs measured with the AirmoVOC (Debevec et al., 2017). Note that the two GCs were deployed at CAO from January 2015 to February allowing for direct comparisons of BVOC levels recorded in March to summertime values.

Additional VOCs were measured at a time resolution of 10 min using an on-line high-sensitivity proton transfer reaction - quadrupole mass spectrometer (PTR-QMS, Ionicon Analytik GmbH, Innsbruck, Austria; Lindinger et al., 1998),

which allowed the detection of protonated OVOCs (alcohols, aldehydes, ketones and carboxylic acids), aromatics (sum of C8 and C9) and BVOCs (e.g. isoprene and the sum of monoterpenes). This instrument has been extensively described in recent reviews (Blake et al., 2009 and references therein) and a description of the analytical setting implemented here and



calibration procedure were given in Debevec et al. (2017). The detection limit of the sixteen protonated compounds typically ranged from 11 to 203  ppt, and relative uncertainty was evaluated between 18 % and 44 % (Debevec et al., 2017).

**Off-line VOC measurements:**

Additionally, more than 400 off-line 3h-integrated air samples were collected on sorbent cartridges (multi-sorbent
and DNPH (2,4-dinitrophenylhydrazine) cartridges), using an automatic clean room sampling system (ACROSS, TERA Environment, Crolles, France). $C_1$-$C_{16}$ organic compounds were sampled for 3 hours via a 0.635 cm diameter 4-m length PFA line and then trapped into one of the two types of cartridges: a multi-sorbent cartridge composed of carbopack C (200 mg) and carbopack B (200 mg) (carbotrap 202, Perkin-Elmer, Wellesley, Massachusetts, USA) and a Sep-Pak DNPH-Silica cartridge (Waters Corporation, Milford, Massachusetts, USA). These techniques are described in Detournay et al.
(2011) and their set up in the field further presented in Detournay et al. (2013) and Ait-Helal et al. (2014). Briefly, thirty-nine $C_5$-$C_{16}$ NMHCs, including alkanes, alkenes, aromatics, nine BVOCs,  along with six $C_6$-$C_{11}$ n-aldehydes, were sampled at a flow rate of 200 mL min$^{-1}$ on the multi-sorbent cartridges preliminary conditioned during 24 h with purified air at 350 °C and 10 mL min$^{-1}$ flowrate, using a RTA oven (French acronym for *"régénérateur d'adsorbant thermique"* - TERA Environment, Crolles, France). Ten additional $C_1$-$C_8$ carbonyl compounds were sampled in parallel with the DNPH
cartridges at a flowrate of 1.5 L min$^{-1}$. During the sampling, different ozone scrubbers have been used in order to avoid any possible ozonolysis of the monitored compounds: a $MnO_2$ ozone scrubber was employed for the multi-sorbent cartridges while KI ozone scrubber was installed upstream of the DNPH cartridges. In addition, stainless-steel particle filters of 2 μm diameter porosity (Swagelok) were used to prevent any sampling of particles. Samples have been later analyzed in the laboratory by GC-FID (with TurboMatrix 650 ATD, Perkin-Elmer, Wellesley, USA; for the multi-adsorbent cartridges) or
HPLC-UV (high-performance liquid chromatography coupling with ultra violet detection; for the DNPH cartridges). The reproducibility of the analysis was checked regularly by the analysis of a standard, leading to the plotting of a control chart for each compound, which allowed the reproducibility of each instrument to be checked. The detection limit of the VOCs measured with off-line techniques was typically below 5 ppt for the multi-sorbent cartridges and ranged from 6 to 27 ppt for the DNPH cartridges. Relative uncertainty was evaluated between 3 % and 26 % for the multi-sorbent cartridges, and
between 11 % and 37 % for the DNPH cartridges (Ait-Helal et al., 2014).

**VOC intercomparison:**

α-Pinene and β-pinene measured by both on-line GC-FID and off-line techniques were selected to cross-check the quality of the results recorded during the campaign. On-line measurements were additionally averaged on a 3-hour time scale to allow direct comparison with off-line measurements and reported in section SI-1 in the Supplement. α-Pinene showed a
better determination coefficient than β-pinene ($r^2$: 0.69 and 0.47 for α-pinene and β-pinene, respectively) and a slope closer to one (1.08 for α-pinene and 0.67 for β-pinene). The results from α-pinene and β-pinene investigate in this paper were taken from the GC-FID measurements due to higher time resolution and better analytical performance of AirmoVOC. Additionally, the sum of six monoterpenes collected by multi-sorbent cartridges was also in comparison to the non-speciated



monoterpenes measured by PTR-MS (section SI-1 in the Supplement), yielding to similar variability and consistent ranges of concentrations ($r^2$: 0.73; slope: 0.79).

Concerning OVOCs, acetaldehyde, acetone and MEK (methyl ethyl ketone) were monitored by both PTR-MS and off-line technique. According to section SI-2 in the Supplement, these OVOCs showed good determination coefficients ($r^2$ of 0.81, 0.90 and 0.84 for acetaldehyde, acetone and MEK, respectively) with slopes close to one for each compound (1.16, 0.87 and 1.04 for acetaldehyde, acetone and MEK, respectively) and relatively low intercepts (77 ppt for acetaldehyde, 86 ppt for acetone and 9 ppt for MEK). Acetaldehyde, acetone and MEK measurements presented in the following are those resulted of PTR-MS by reason of a finer time resolution.

As a consequence, recovery of the different techniques, frequent quality checks and uncertainty determination approach have allowed to assure a satisfying robustness of the dataset and cross check comparisons have shown comparable results for the different techniques used (within the range of uncertainties).

### 2.2.2 Ancillary gas measurements

A large set of real-time atmospheric measurements was performed by the DLI at the CAO, in order to characterize trace gases (NO, $NO_2$, $O_3$, CO and $SO_2$). These latter are presented in more details by Kleanthous et al. (2014). The time resolution was 5 min for each analyzer. The results examined in this study are hourly average.

### 2.2.3 Aerosol measurements

Particle size distribution measurements were performed using a setup of a custom-made differential mobility particle sizer (DMPS, TSI Inc., model 3080; Villani et al., 2008) completed by a particle-size magnifier (PSM, Airmodus, model A09; Vanhanen et al., 2011). The DMPS consists of a bipolar charger to charge the aerosol particle population to the equilibrium charge distribution, a 28-cm differential mobility analyzer (DMA) in a close sheath-air loop and a condensation particle counter (CPC, TSI Inc., model 3010). This instrument was operated to measure the aerosol size distribution over 20-200 nm diameter size range from 8 to 11 March, and over the 10-250 nm size range from 12 to 27 March, with a time resolution of 460 s. $N_{DPMS}$ was used in this paper to refer to total number concentrations of particles obtained by integrating the DMPS measurements. Total number concentrations of particles larger than 1 nm diameter ($N_{PSM}$) were measured with a PSM using diethylene glycol (DEG) as the working fluid at a fluid flow rate of 1 standard liter per minute. A PSM can grow particles as small as 1 nm to larger than 90 nm, after which a CPC is used to count the grown particles. Considering its time resolution (1 s), PSM data were filtered from local pollution spikes due to the local anthropogenic activity on the CAO site (between 07:00-17:00 local time (LT) during week days). Finally, the particle cluster and sub-10 nm particle (between 1 and 10 nm) concentrations were calculated as the difference between the total particle concentration derived from the DMPS and the PSM concentrations ($N_{PSM}$ - $N_{DPMS}$).

The charged cluster size distributions were recorded with a Neutral cluster and Air Ion Spectrometer (NAIS). This spectrometer is a modified version of the AIS instrument (Airel Ltd, Mirme et al., 2007; Mirme and Mirme, 2013) which is





an instrument capable of measuring mobility distributions of sub-3 nm charged aerosol particles and clusters. Controlled charging, together with the electrostatic filtering, enables it to additionally measure the neutral aerosol **p**articles distribution. The measurement principle of the NAIS is based on two independent spectrometer columns, one of each polarity, where the ions are classified by a DMA. More details are given in Manninen et al. (2011). The mobility range is 3.2-0.0013 $cm^2.V^{-1}.s^{-1}$,

corresponding to particle Milikan diameter between 0.8 and 42 nm.

     The chemical composition of non-refractory submicron aerosol (NR-PM$_1$) has been non-stop monitored by deploying a quadrupole aerosol chemical speciation monitor (Q-ACSM, Aerodyne Research Inc., Billerica, Massachusetts, USA), which has been fully characterized by Ng et al. (2011). This instrument shares the same general structure with the aerosol mass spectrometer (AMS) aside from it has been specifically thought for long-term monitoring purposes. The Q-

ACSM instrument was operating continuously with 30-min time resolution during the whole duration of the campaign totalizing 1292 valid data points (corresponding to a time recovery of 95 %). The ACSM dataset was validated by comparison with co-located PM$_1$ chemical composition results obtained by integrated daily (24 h) time resolution filter based measurements. Instrument settings, field operation, calibration and data processing are those reported in Petit et al. (2015).

     Black carbon (BC) was calculated using the 880 nm channel of a 7-wavelength (370, 470, 520, 590, 660, 880 and

950 nm) Aethalometer (AE31 model, Magee Scientific Corporation, Berkeley, CA, USA) with a time resolution of 5 min. Presuming difference in the absorption angstrom exponent between fossil fuel and biomass burning derived aerosol, the BC originating from these two sources was apportioned following the method described by Sandradewi et al. (2008).

**2.2.4 Meteorological measurements**

Meteorological parameters (temperature, pressure, relative humidity, wind speed, wind direction and radiation) were

monitored every 5 min using a weather station (Campbell Scientific Europe, Antony, France) located on the roof top of the CAO building, at approximately 5 m a.g.l.

     Classification of air-mass origins has been based on the analysis of the retroplumes computed by the Flexpart lagrangian model (Stohl et al., 2005) considering CAO as the receptor site. The Flexpart model simulates trajectories of user-defined ensembles of particles released from three-dimensional boxes. The classification was based on hourly resolution

model simulations going back in time to 5 days, taking into account only the lowest 100 m a.g.l. (footprint plots), even if the 3 km was modeled. These backward retroplumes were classified within 8 source regions, similar to Kleanthous et al. (2014), identified by a custom-made algorithm combined with visual inspection. The source region map is depicted in Fig. 2 based on the residence time of particles over each source region. During March 2015, the CAO station was mostly under the influence of continental air masses originating from "Southwest Asia" (cluster 7 – 31 %), "Northwest Asia" (cluster 4 –

28 %), "West of Turkey" (cluster 5 – 10 %) and "Europe" (cluster 3 – 11 %) together with by marine air masses (cluster 2 – 14 %). Note that, air masses categorized as "local" (cluster 0) occurred only on 23 and 24 March and may rather be considered as a transitory state between periods of air masses originating from Northwest Asia and West of Turkey. It is



worth noting that March 2015 was characterized by an unusually high contribution of Southwest Asian air masses in the detriment of European air masses compare to the period 1997-2012 investigated in Kleanthous et al. (2014).

## 2.3 Identification and contribution of major sources of VOCs

A source apportionment using positive matrix factorization (PMF) was conducted in Debevec et al. (2017) to better

determine covariance factors of VOCs representative of aged or local primary emissions as well as secondary photochemical transformations taking place during the air mass transport. The US EPA PMF v. 5.0 was applied to the 30-min time resolution March 2015 dataset composed by 20 VOCs (including OVOCs measured on-line) and a total of 1179 atmospheric data points. As results from this PMF study will be partly used in this study, a short description of the corresponding results is given here.

10        The best PMF solution allowed the deconvolution of measured VOCs into six distinct factors. Factors imputed to biogenic sources 1 and 2 (relative contribution of 43 % to the total mass of VOCs), driven by pinenes and isoprene/OVOC emissions, respectively, have shown contrasted diurnal profiles (nighttime vs. daily maxima) and were assigned as originating from different types of emitting vegetation (oak and pine forests vs. garrigues). Factors imputed to anthropogenic sources (short-lived combustion source, evaporative sources, industrial and evaporative sources, 21 % altogether) were

characterized by compounds of various lifetimes and were identified either of local or regional origins. The last factor (36 %) was characterized by long-lived primary anthropogenic VOCs and OVOCs and covaried with CO, supporting its identification as continental regional background. Chemical profile, variability and origin of these factors are discussed with more details in Debevec et al. (2017).

## 2.4 Evaluation of properties for new particle formation events

### 2.4.1 Particle formation and growth rates calculations

The most relevant variables for identifying NPF events are the formation rate ($J_i$ expressed in $cm^{-3}s^{-1}$) at a given diameter (i, in nm) and the growth rate (GR, in nm.h$^{-1}$), which is defined as the diameter rate of change due to particle population growth. The growth rate between two size classes was calculated considering the method defined by Hirsikko et al. (2005) which is based on the time corresponding to the maximum concentration in each size class of the selected size range by

fitting a normal distribution to the size class concentration.

Formation rates were especially evaluated for the very first steps of the formation process, i.e. between 1 and 3 nm (Kontkanen et al., 2017). As previously mentioned, the PSM was measuring in a scanning mode during the studied period, but the differences between the concentrations of the successive size classes were too small to allow determination of size distributions, and hence any growth rate calculation. The total particle formation rate was thus calculated at 1.5 nm ($J_{1.5}$)

from the total particle concentration measured in the size range $1 - 2.5$ nm by the PSM ($N_{1-2.5}$), by using the growth rate in the size range $1.5 - 3$ nm ($GR_{1.5-3}$ in nm.h$^{-1}$), and the loss of particle by coagulation scavenging of 1.5 nm particles on



larger pre-existing particles ($CoagS_{1.5}$ in $s^{-1}$) both derived from the NAIS measurements. The growth rate of the corresponding size range is then obtained by a linear least square fit through the time values previously found. The total particle formation rate at 1.5 nm was finally calculated according to Eq. (1), from Kulmala et al. (2007):

$$J_{1.5} = \frac{dN_{1-2.5}}{dt} + CoagS_{1.5} \times N_{1-2.5} + \frac{1}{1.5 \, nm} GR_{1.5-3} \times N_{1-2.5} \qquad \text{(Eq. 1)}$$

### 2.4.2 Condensation sink

CS denotes the ability of the particle size distribution to remove condensable vapor from the atmosphere and hence describes the loss rate of the condensable vapors onto the pre-existing particles (Pirjola et al., 1999). This variable is proportional to the surface area density of an aerosol particle and has been calculated based on size distribution measured with DMPS as proposed by Kulmala et al. (2001).

### 2.4.3 Sulfuric acid

$SO_2$ produces ambient $H_2SO_4$, which is currently thought to be the most likely nucleation precursor candidate as well as contributes to the growth of newly formed particles (Kulmala et al., 2013; Sipilä et al., 2010). To study the connection between NPF and $H_2SO_4$, an empirical proxy for $H_2SO_4$ concentration was calculated from the $SO_2$ concentration according to Eq. 2, adapted from Mikkonen et al. (2011) which is based on previous work by Petäjä et al. (2009):

$$[H_2SO_4]_{calc} = 2.468.10^{-11} \times \frac{GlobRad \times [SO_2]^{1.385}}{(CS \times RH)^{1.03}} \qquad \text{(Eq. 2)}$$

where, GlobRad is the global radiation in $W.m^{-2}$, $[SO_2]$ is the sulfur dioxide concentration in $molec.cm^{-3}$, CS is the
condensation sink in $s^{-1}$ and RH is the relative humidity. The coefficients used in Eq. 2 were calculated from sulfuric acid measured with a CIMS instrument (Sellegri et al., 2016). This proxy was constructed for radiations higher than 10 $W.m^{-2}$ but the predictive ability is significantly raised for radiations exceeding 50 $W\,m^{-2}$ (Rose et al., 2015).

### 3 Results and discussions

### 3.1 General overview of ambient BVOC levels

### 3.1.1 Ambient concentration levels

Nine BVOCs, namely α-/β-pinenes, α-/γ-terpinenes, limonene, myrcene, camphene, 3-carene and isoprene, have been detected and quantified at the CAO and their mean levels during March 2015 are presented in Fig. 3. The average concentration of the sum of terpenoids during March 2015 was 282 ± 307 ppt. Among BVOCs, the most abundant were monoterpenes. The average concentration of monoterpenes during the intensive field campaign was 236 ± 294 ppt with
maximum up to 4500 ppt (recorded during the night of 10 March). Monoterpenes exhibited high daily amplitude, with a mean mixing ratio of 154 ppt during the daytime time against 329 ppt during the nighttime (Fig. 3). Higher





concentrations of monoterpenes (estimated by the concentrations of the sum of α-pinene and β-pinene) were observed during the summertime (307 ppt on average – Fig. 3) but maximum concentrations were at the same order of magnitude (e. g. a peak up to 3600 ppt was recorded during the night of 31 July – not shown here). The dominant monoterpenes observed during the field campaign were β-pinene (79 ± 106 ppt) and α-pinene (73 ± 78 ppt) followed by

limonene (27 ppt), camphene (25 ppt), $\Delta^3$-carene (11 ppt), myrcene (6 ppt), α-terpinene (3 ppt) and γ-terpinene (below 1 ppt). α-Pinene and β-pinene accounted together for 67 % of the total monoterpenes concentration. Average concentration of isoprene was quite low (46 ppt) in March 2015 but it was higher by a factor of 3 in the summertime (Fig. 3) due to higher temperatures. As a matter of fact, isoprene, α-pinene and β-pinene are the major BVOCs emitted by the Mediterranean vegetation (Owen et al., 2001). In addition to their high emission rates by vegetation, they are the less reactive

isoprenoids with OH radicals and ozone (Atkinson and Arey, 2003) and therefore tend to accumulate (for short periods) in the atmosphere. Other more reactive compounds, such as α-terpinene and limonene, are removed very quickly following their emission, thus exhibit lower concentrations in the atmosphere.

An overview of BVOC concentrations at different background locations in the Mediterranean is depicted in Fig. 3. As for CAO, Cape Corsica and Finokalia are representative remote sites with Mediterranean shrublands and primary BVOC

concentrations recorded at these sites have similar seasonal behaviors and concentration levels. Speciated monoterpenes measured at remote/rural Mediterranean sites were predominantly composed of α-pinene and β-pinene with a higher proportion of camphene and 3-carene observed only at CAO. Contrarily, α-terpinene was observed in higher proportion during the summer field campaign performed at Cape Corsica.

### 3.1.2 Temporal variability and sources

As shown in Fig. 4, the diurnal variations of isoprene and monoterpenes present opposite diurnal evolutions. Daily amplitude is of 317 ppt on average for monoterpenes, with low during daytime hours, a significant increase after sunset (17:00-18:00 LT), high concentrations throughout the night, and decreasing after sunrise (06:00 LT). The monoterpene average diurnal patterns indicated that their emissions were solely dependent on temperature (Geron et al., 2000a and references therein) and lower, but still significant emissions occurred throughout the night. A similar pattern with nighttime maxima has been

observed at other locations in the Mediterranean (e. g. in Portugal by Cerqueira et al., 2003; in France by Detournay et al., 2013; in Italy by Kalabokas et al., 1997 and Davison et al., 2009; and in Greece by Harrison et al., 2001) and was assigned to nocturnal emissions of monoterpene stored in the understorey vegetation (Niinemet et al., 2004; Schurgers et al., 2009). These nighttime maxima are enhanced by the low removal processes (i.e. low oxidizing species concentrations) and the shallow nocturnal boundary layer which concentrate close to ground level the monoterpenes emitted by vegetation.

Furthermore, the prevalent nocturnal winds at CAO (originating from the Southwest to Southeast sectors) may have also contributed to these nighttime maxima with air masses enriched with biogenic emissions from the forests located in the Troodos Mountains (Debevec et al., 2017). The major vegetation types covering these mountains are pine forests, composed of Calabrian pines (*Pinus brutia* – from foothills to the high mountains up to 1200 m) and black pines (*Pinus nigra* – on the



highest peaks at altitudes from 1400 m to 1951 m); and oak forests, mostly composed of golden oaks (*Quercus alnifola* – found with *Pinus Brutia* or in inland maquis between about 800 and 1,500 m elevation) and Kermes oaks (*Quercus coccifera* – up to 1,400 m elevation) (Fall, 2012 and references therein). These coniferous species are considered as very strong emitters of monoterpenes (Aydin et al., 2014a, 2014b). More specifically, oaks are usually classified as predominant

isoprene emitters (Helmig et al., 2013), whereas *Quercus coccifera* are considered as significant emitters of α-pinene and β-pinene in Owen et al. (2001). A considerable fraction of the monoterpene emission from pines originates from large storage structures (Niinemets et al., 2004) and, as such, continue to emit at a higher relative level during the nighttime compared to oaks as long as nocturnal temperatures remain sufficiently high (Laothawornkitkul et al., 2009; Owen et al., 1997).

With daily amplitude of isoprene of 43 ppt on average, the observed isoprene pattern followed an usual diel profile controlled by temperature and solar radiation (Geron et al., 2000b; Owen et al., 1997). As depicted in Fig. 4, isoprene concentrations started to increase immediately at sunrise (06:00 LT), indicative of local biogenic emissions. However, isoprene concentrations did not decrease immediately at sunset (17:00 – 18:00 LT) but remained rather constant at the beginning of the night (considering upper end of the whiskers and mean values of hourly box plots depicted in Fig. 4) and

followed a slow decrease until reaching a minima (03:00 LT). Isoprene levels showed their minimum levels during the night, although levels up to 200 ppt could be noticed during the night of 10 March and coincided with the highest concentrations of monoterpenes recorded during the field campaign. This finding suggests that air masses were enriched with biogenic emissions, the main contributors of monoterpenes, were also partially isoprene emitters as reported previously (Detournay et al., 2013). This finding is also in agreement with the source apportionment reported in Debevec et al. (2017). Two VOC

biogenic sources were identified, they were both composed of different primary biogenic species (pinenes and isoprene for factor 1 and factor 2, respectively) and showed distinct temporal variabilities and geographic origins supporting the division of BVOC sources into two factors. Biogenic factor 1, driven by pinenes emissions, also explained a small proportion of isoprene (15 %, Debevec et al., 2017). Nonetheless, the contribution of 2-methyl-3-butene-2-ol (MBO) to the isoprene signal of PTR-QMS m/z 69 cannot be discarded (Karl et al., 2012; Kim et al., 2010). Isoprene usually dominates over most other

BVOCs in many places and these interferences are often shown to be minor (Karl et al., 2004; Misztal et al., 2011; Warneke et al., 2010). However, measurements, particularly in coniferous ecosystems, can be of greater analytical challenge due to the concomitant emission of isoprene and MBO (Kim et al., 2010; Schade and Goldstein, 2001). In contrast to monoterpenes, the emissions of MBO require light as isoprene. Isoprene and/or MBO that were emitted during the late afternoon could be not fully oxidized photochemically, as OH concentrations begin to fall, and could remain in the nighttime atmosphere.

**3.2 Factor controlling BVOC emissions**

In this section, time variations of main monoterpenes and isoprene are examined along with meteorological parameters in order to determine the dominant emission drivers for BVOCs. Five events are highlighted for that purpose in Fig. 5, and correspond to periods when elevating mixing ratios of pinenes (higher than 500 ppt) were observed. Pinene variations during



moderate events (i. e. events 1 2, 4 and 5 in Fig. 5) will be discussed in order to finally understand why such elevated mixing ratios of monoterpenes were observed during the night of 10 March (event 3 in Fig. 5).

Isoprene and monoterpene emissions are known to be controlled by ambient temperature (Guenther et al., 2000). Consistently, high isoprene concentrations were noticed during the warmest days of the campaign (8-10 March) with a

maximum temperature of 26 °C. A closer look at event 2 (i. e. 8 March) shows that pinene concentrations were spiked up to 800 ppt, a value which is much higher than those observed during the previous night. As expected, pinene emissions were enhanced by an increase of ambient temperatures since maximum temperature recorded during event 2 was 6 °C higher than the one of the previous day. This dependency to temperature is consistent with the previous discussion related to monoterpene daily variations.

At CAO, significant changes in monoterpene mixing ratios appeared to occur during and after rainy periods. This phenomenon was observed during event 4 (i. e. 11 March) when high levels of pinenes (up to 800 ppt) were observed during daytime rainfall but also after this episode, although temperatures during event 4 (i. e. 12-13 °C) were among the lowest of the month. A rainy period was also noticed in the morning of event 5 (i. e. 28 March) and corresponded again to a pinene peak of 800 ppt. Pinene concentrations during the following night were a factor of 3 higher compared to mixing ratios at

similar temperature and relative humidity (e. g. 24 March). These results suggest that, rainfall has induced a stress factor onto the vegetation and therefore may have caused short-term increases in the release of monoterpenes from the vegetation. This assumption is backed by results from plant enclosure experiments (Lamb et al., 1985) as well as several field measurements (Bouvier-Brown et al., 2009; Davison et al., 2009; Helmig, 1999; Schade et al., 1999). Additionally, rainy periods are also usually characterized by low OH concentrations which could promote the accumulation of BVOCs in the

atmosphere. Furthermore, the stimulation of pinene emissions by rainfall seemed to be responsible for the significant monoterpenes concentrations observed during the daytime. The influence of precipitation on isoprene emissions was not clearly identified here.

Monoterpene emissions at CAO could be more strongly dependent on humidity than temperature under dry conditions. Higher pinene concentrations (up to 1,100 ppt) were observed during event 1 (i. e. 3 March) compared to pinene

levels recorded on the days before and after event 1. An increase of relative humidity of 20 % seemed to be sufficient to induce higher pinene concentrations of a factor 3 compared to mixing ratios observed on 2 and 4 March at similar temperature (maximal temperatures of 16-17 °C) but lower relative humidity. Additionally, pinene concentrations during event 1 were slightly higher than those of event 2 while temperature and relative humidity were significantly different these days. Indeed, temperatures of event 1 were lower than temperatures of event 2 (maximal temperature of 16 °C and 22 °C for

event 1 and event 2, respectively) which would seem to be compensated by higher relative humidity during event 1 compared to ones of event 2 (up to 90 % and 65 % for event 1 and event 2, respectively). Humidity has also been found to increase monoterpene emission rates (Janson, 1992, 1993; Lamb et al., 1985; Schade et al., 1999). These studies pointed out that monoterpene emissions rates correlated with relative humidity because wet needle surfaces emit greater absolute amounts and different relative amounts of terpenes than dry needles. Additionally, lower boundary layer heights are



generally observed on non-sunny days compared to sunny days ones, this would enhancing monoterpene maxima by the shallow nocturnal boundary layer. Nocturnal concentrations of isoprene would seem to usually occur at a high relative humidity.

Looking finally at event 3, ambient mixing ratios of pinenes and isoprene were the highest ones observed during the intensive field campaign and event 3 is among the warmest and most humid periods of the campaign. Pinene concentrations during event 3 were a factor 4 higher compared to concentrations observed during events 1 and 2. Nocturnal fog could be an assumption as reported by Janson (1993), who noticed high monoterpene emission rates during the nighttime when a radiation fog developed in an experiment chamber, inducing high relative humidity (> 90 %).

As a summary, BVOC emissions have shown to be controlled by ambient temperature, precipitation and relative humidity. More specifically, significant increases in monoterpene mixing ratios occurred during and after rainy periods and the stimulation of pinene emissions by rainfall seemed to be responsible for additional emissions of monoterpenes during the daytime. High relative humidity seemed to promote high BVOC concentrations originating from the nocturnal biogenic source (i. e. oaks and pines forests).

## 3.3 OVOC sources

In addition to isoprene and monoterpenes, several OVOCs can be emitted by plants. Six OVOCs have been detected and quantified at CAO by on-line instrumentation. With an average concentration of 4703 ± 2224 ppt, these six OVOCs represented a high fraction of the total concentration of VOCs measured in March 2015 (Debevec et al., 2017). The dominant OVOCs observed during the field campaign were those with the higher lifetimes, i. e. methanol (2875 ± 1589 ppt, 12 days - Debevec et al., 2017) and acetone (1125 ± 398 ppt, 68 days), followed by acetaldehyde (452 ppt, 19 h), MEK (221 ppt, 9 days) and MVK+MACR (32 ppt – 10-14 h). Off-line instrumentation also provided formaldehyde with an average concentration of 986 ppt (29 h).

OVOCs can be either emitted from primary sources (mainly biogenic) or be produced by secondary sources related to the oxidation of anthropogenic and biogenic VOCs, making more complicated to assess their origins. From the 6 PMF factors reported in Debevec et al. (2017), the measured OVOCs were distributed among their different sources (Fig. 6). More than 80 % of the respective total mass of methanol, acetaldehyde and MVK+MACR was explained by biogenic sources, especially by factor 2 driven by isoprene emissions. Acetone and MEK were mainly attributed to local biogenic sources and to more distant sources. However, the PMF analysis did not allow to distinctly deconvoluate primary sources from secondary ones. On the other hand, even if isoprene and its first oxidation products (MVK+MACR) were both included in factor 2, a delay of about 1 hour in the peak values could be observed between isoprene and its first oxidation products (Fig. 7) making possible to separate primary from secondary contributions of factor 2.

Based on these results, methanol and acetaldehyde temporal patterns were further explored in the light of the variabilities of isoprene and its oxidation products in Fig. 7. Based on the budget estimation reported by Jacob et al. (2005), methanol is likely to be dominated by biogenic emission sources resulting from the demethylation of pectin during plant cell



wall expansion (Galbally and Kirstine, 2002; Hüve et al., 2007). Another important source is the photochemical production from methane under very low $NO_x$ conditions (Schade and Goldstein, 2006). The methanol pattern observed at CAO followed a typical diel profile and correlated quite well with temperature variation ($r^2 = 0.49$). As depicted in Fig. 7, methanol concentrations started to increase immediately at sunrise (06:00 LT). The morning increase pattern of methanol

concentration is similar to isoprene one, suggesting a primary biogenic source as reported elsewhere (e. g. Karl et al., 2001, 2003; Schade and Goldstein, 2001). Studies have shown that methanol can build up within the stomata during the night, releasing a large burst to the atmosphere when the stomata open (morning bursts), followed by emissions consistent with changes in stomatal conductance (Hüve et al., 2007) and temperature (Harley et al., 2007). This is consistent with the observed measurements of increasing concentrations in the early morning, coincident with stomatal opening.

Hydrocarbon oxidation (mostly alkanes and alkenes but also isoprene and ethanol) provides the largest acetaldehyde source in the budget estimates of Millet et al. (2010). Nonetheless, for all reaction pathways of isoprene with atmospheric oxidants, acetaldehyde is produced as a second- or higher-generation oxidation product of isoprene (Millet et al., 2010). In addition to photochemical production, acetaldehyde is emitted by terrestrial plants, as a result of fermentation reactions leading to ethanol production in leaves and roots (Jardine et al., 2008; Rottenberger et al., 2008; Winters et al., 2009). Along

with methanol and isoprene, acetaldehyde showed a similar trend with a clear diurnal cycle with a daytime maximum consistent with temperature. Acetaldehyde peaked at midday followed by a gradual decrease throughout the rest of the day. Isoprene and acetaldehyde also correlated well ($r^2 = 0.49$) suggesting acetaldehyde was mostly released in the atmosphere by local vegetation.

## 3.4 Impact of BVOCs on nucleation and NPF events

### 3.4.1 NPF event identification and classification

The days during the measurement period were classified with respect to whether or not new particle formation (NPF) was observed. The NPF event days were identified using PSM and DMPS measurements and based on the criteria and methodology reported by Dal Maso et al. (2005). Briefly, a day was classified as an NPF event if (1) a clear increase in the fine particulate mode was observed, (2) followed by a sustained growth for at least a couple of hours until reaching a

relevant particle size to form cloud condensation nuclei (CCN), resulting in a well-known "banana shape" as depicted in Fig. 8. Out of 20 observation days, such events were observed on 14 days (i. e. 8-10, 14, 16-18, 20-23 and 25-27 March) during daytime. The beginning of each event corresponds to the time when a new nucleation mode appeared, and its end is defined as the stop of the subsequent growth. On most NPF event days (8-10, 14, 16, 18, 23 and 25-27 March), we observed a steep increase of particle number (i. e. $N_{DMPS}$ increased up to 25000 particles.$cm^{-3}$) initiated during the morning. On each

NPF event day, a clear increase of the nanoparticle concentration, calculated as the difference between $N_{PSM}$ and $N_{DMPS}$ was observed prior to the $N_{DMPS}$ number concentration increase. This indicates that the NPF events observed at CAO are initiated in the vicinity of the measurement site at the same time than at the regional scale, as the growth of these clusters are



measured continuously over several hours as they are transported from further distance from the measurement site. On the 17, 20 and 22 March, cluster concentration increases were observed, indicating that nucleation was occurring in the local environment, but they were not followed by newly formed particle growth, and hence not observed at the regional scale. The 6 remaining days out of the 20 observation days were classified as non-event days (i. e. 11-13, 15, 19 and 24 March).

5        The event days were classified further into subclasses (Ia, Ib, II and apple) according to the classification by Yli-Juuti et al. (2009) based on previous work by Hirsikko et al. (2007) and Vana et al. (2008). This classification depends on the event applicability to growth and formation rates analysis. Class I presents the days when the formation and growth rate are determined with a good confidence level. Class I is divided into Class Ia and Ib. The Class Ia event (14, 18, 23 March) has clear and strong particle formation with little or no pre-existing particles, while a Class Ib event is any other Class I event

(8-10, 20 and 25 March) where the particle formation and growth rate still can be determined. The Class II event (16-17 and 22 March) represents the event where the accuracy of formation rate calculation is questionable due to data fluctuation even though the banana shapes are still observable. Class apple event (21 and 26-27 March) refer to an increase in the fine particulate mode, but newly formed particles do not show clear growth and hence there is a clear gap between the new mode and other modes. Note that particulate formation and growth rates discussed in this study were only calculated for Ia and Ib

events.

### 3.4.2 Overview of factors influencing nucleation events

To investigate the factors governing daytime NPF processes, it was decided to partition the NPF days in 4 classes (NPF1-NPF4) based on prevailing atmospheric conditions. Time series of various tracers are shown in Fig. 9. This classification was based on the transport pathways of the air masses arriving at CAO site and mean daily concentrations of these

parameters depicted in Fig. 10 as mean daily values of properties indicators for NPF events (i. e. CS, particulate formation and growth rates). Figure 11 shows diurnal cycles of parameters with suspected influence on NPF, averaged over event and non-event days. Note that, no clear trend was observed on 21 March.

**Classification of NPF days as a function of atmospheric conditions:**

       NPF1 event days (i. e. 18 and 25-27 March) and NPF2 event days (i. e. 8-10 and 23 March) mainly concerned class

I and apple type events. The CAO station may receive pollution of local/regional origins since winds during NPF1 and NPF2 event days were mainly of Northeastern, Eastern and Southeastern directions and the station received air masses originating from Southwest and Northwest Asia (Fig. 9). In the course of these NPF event days and especially for NPF2 ones, high concentrations of $PM_1$, HOA, BC and $NO_2$ (Fig. 10), which are recognized as tracers of anthropogenic sources, suggested that NPF events were of anthropogenic origin. High concentrations of inorganic compounds ($SO_4$, $NO_3$ and $NH_4$) were also

observed suggesting the air masses sampled during NPF1 and NPF2 events were both polluted and aged. Additionally, higher BVOC concentrations (i. e. isoprene, MVK+MACR and monoterpernes – Fig. 10) were observed during NPF2 event days than the ones during NPF1. Consequently, NPF1 event days were characterized mainly of anthropogenic origin while NPF2 event days were of mixed origins (anthropogenic and biogenic).




NPF3 event days (i. e. 14, 16-17 and 22 March) and NPF4 event days (i. e. 20 March) concerned all class II events and some class I events. During these NPF event days, the CAO station received winds which were mainly of Northwestern, Western and Southeastern directions and the station was mainly influenced by maritime air masses (Fig. 9) and continental ones which have not been newly in contact with anthropogenic sources. This finding was confirmed by the low

anthropogenic tracer concentrations observed during these NPF days which were similar to the ones observed during non-event days. Moreover, BC didn't exceed 0.5 µg.m$^{-3}$ supporting the consideration of atmospheric conditions as clean conditions according to Cusack et al. (2013). Higher isoprene concentrations than the ones characterizing non-event days were only noticed during NPF4 event even if MVK+MACR mean concentrations of NPF4 event days were slightly lower than the one observed on the non-event days (Fig. 10). As a result, NPF3 event days were characterized mainly of marine

origin while NPF4 event days were probably of biogenic origin. Note that PM$_1$ concentrations during NPF4 event days were of the same range as the ones recorded during NPF event days of anthropogenic origin (Fig. 10) due to higher concentrations of inorganic compounds (i. e. SO$_4$ and NH$_4$) and a higher contribution of Low Volatile Oxygen-like Organic aerosol (LV-OOA) to OM concentration. These findings suggest highly processed (aged) regional background pollution transported to the receptor site on NPF4 days.

**Study of the effect of meteorological parameters on new particle formation:**

The intensity of global solar radiation was higher on event days (282 W.m$^{-2}$ in average) than on non-event days (204 W.m$^{-2}$). This finding was consistent with the literature (Cavalli et al., 2006; Cusack et al., 2013; Guo et al., 2008; Hamed et al., 2007). Solar radiation is known as an important parameter in the initial step of atmospheric nucleation, since photochemical reactions among various chemicals were facilitated by stronger solar radiation leading to the production of

the nucleating and/or condensing species involved in NPF (Harrison et al., 2000).

The hourly-average RH followed the opposite temporal pattern as that of the intensity of global radiation. The RH was hence lower on event days (61 % in average) than on non-event days (80 %) as observed elsewhere (Boy and Kulmala, 2002; Guo et al., 2012; Hamed et al., 2007). This could be partly explained by the fact that lower RH days usually have less clouds causing more solar radiation and subsequently producing more OH radicals to form more condensable vapors (Hamed

et al., 2007). Another possible reason could result from an increase in particle hydration with increasing RH, which leads to larger pre-existing particle surface areas and thereby larger condensation sinks, resulting in a possible inhibition of the nucleation (Birmili et al., 2003; Hamed et al., 2011).

Temperatures were also higher on event days (i. e. NPF1-NPF3: 13.8 °C in average) than on non-event days (11.2 °C) and especially during NPF1 event days (15.4 °C). As for RH, higher temperatures might be linked to higher solar

radiation associated to higher photochemistry, and it is difficult to separate the two effects with our datasets. Higher temperatures have been associated with the nucleation events in Germany (Birmili et al., 2003), in Italy (Hamed et al., 2007) and in Atlanta (Woo et al., 2001) which may be induced by higher BVOC emissions (Guo et al., 2008). Contrarily, lower temperatures have been associated with the nucleation events in Finland (Boy and Kulmala, 2002; Vehkamäki et al., 2004)



and in Hong Kong (Guo et al., 2012). These later findings could be due to lower temperature at the start time of the NPF events may enhance the nucleation of $H_2SO_4$ with water vapor (Guo et al., 2012).

**NPF1 and NPF2 event days: nucleation under polluted atmospheric conditions**

NPF1 and NPF2 event days occurred at CAO at high levels of $SO_2$ (both at 0.7 ppb in average and up to 2.2 ppb - Fig. 10-11) compared to $SO_2$ ones on non-event days (0.2 ppb in average and up to 0.4 ppb) suggesting its implication in nucleation formation. Conditions seem to be even more favorable for $H_2SO_4$ production during NPF2 event days since higher $H_2SO_4$ concentrations were observed on NPF2 event days (1.4 $10^8$ molec.cm$^{-3}$ in average – Fig. 11) than on NPF1 event days (6.3 $10^7$ molec.cm$^{-3}$). NPF1 and NPF2 also occurred at high CS (up to 0.4 s$^{-1}$ – Fig. 9) compared to non-event days (up to 0.2 s$^{-1}$). High condensation sink might suppress the particle formation and the growth of the newly formed particles as substantial fraction of the vapors could be condensing on the larger particles. Furthermore, the condensation sink is proportional to the coagulation sink of nucleation mode particles on pre-existing particles. Therefore, at high CS, a high growth rate is required for the newly formed particles to survive and grow to larger sizes instead of being scavenged by coagulation (Kulmala et al., 2005). Thus, for NPF1 and NPF2, a high condensable source rate has to compensate for the high CS.

What further distinguished NPF2 from NPF1 was that isoprene concentrations were particularly high during NPF2 event days (Fig. 10-11) than contributions during non-event days due to more favourable temperatures (Fig. 11). Similar diurnal variations were also observed between isoprene, temperature and $N_{DMPS-PSM}$ during NPF2 event days suggesting that isoprene and $H_2SO_4$ can both play a role during NPF2 event days. Contrary, isoprene concentrations during NPF1 event days were similar to those observed during non-event days that would suggest that NPF1 was mainly induced by $H_2SO_4$.

Slightly lower $PM_1$ concentrations were measured during NPF1 event day (9.7 µg.m$^{-3}$) compared to $PM_1$ concentrations during NPF2 event days (12.9 µg.m$^{-3}$) since lower OM concentrations were observed on NPF1 event days (4.3 vs. 6.8 µg.m$^{-3}$ during NPF1 and NPF2 event days, respectively - Fig. 10) while similar concentrations of $SO_4$ and $NH_4$ were noticed during these NPF event days (2.9-1.9 µg.m$^{-3}$ and 3.3-2.1 µg.m$^{-3}$ for $SO_4$-$NH_4$ concentrations during NPF1 and NPF2 event days, respectively). Moreover, Semi Volatile Oxygen-like Organic aerosol (SV-OOA) contributed to OM more intensively only during NPF2 event days (up to 86 % - Debevec et al., 2017) while LV-OOA contributions were in the same range during NPF1 and NPF2 event days (1.7-1.8 µg.m$^{-3}$). As a conclusion, isoprene contribution during NPF2 event days coincided with higher $PM_1$ concentrations due to a higher contribution of SV-OOA.

**NPF3 and NPF4 event days: nucleation under clean atmospheric conditions**

NPF3 and NPF4 event days occurred at low levels of $SO_2$ (0.3 and 0.2 ppb for NPF3 event days and NPF4 event days, respectively) inducing low $H_2SO_4$ concentrations (Fig. 10) that could be not sufficient to initiate nucleation formation. These NPF events started at low CS (0.05 s$^{-1}$) highlighting a lower uptake of species compounds by condensation and favoring nucleation processes under clean atmospheric condition. Note that, daily CS observed during NPF4 event day were high compared to CS during NPF3 event days and non-event days (Fig. 10) but CS were below 0.1 s$^{-1}$ when NPF4 event was




initiated as depicted in Fig. 9. Additionally, the increase of CS followed the increase of $N_{\text{DMPS-PSM}}$ suggesting that the CS time variation was mainly driven by the growth by condensation of newly formed particles.

A steep increase of particle number $N_{\text{DMPS-PSM}}$ up to 70000 particles.cm$^{-3}$ was recorded between 08:00-10:00 LT on 20 March (i. e. NPF4 event day) when solar radiation and temperature were observed to be intense. Isoprene concentrations increased since 06:00 on 20 March and remained relatively high (60-110 ppt) during the morning, suggesting a role in NPF formation. Contrarily, isoprene concentrations recorded during NPF3 event days were lower than NPF4 event day and non-event days suggesting that a marine source could be involved in NPF3 events.

**NPF2 event days of mixed origins vs. NPF1/NPF4 of individual origin (anthropogenic/biogenic)**

On 23 March (mixed NPF2 event type), the NPF event occurred at similar levels of biogenic tracer (isoprene concentrations between 60-110 ppt – Fig. 9) than the ones observed on 20 March (biogenic NPF4 event type) while higher mean particulate formation and grow rates were met during the selected NPF2 event ($J_{1.5}$: 8.97 cm$^{-3}$s$^{-1}$ - $GR_{1.5-3}$: 3.18 nm.h$^{-1}$) compared to the mean rates characterizing the NPF4 event day ($J_{1.5}$: 8.13 cm$^{-3}$s$^{-1}$ - $GR_{1.5-3}$: 1.93 nm.h$^{-1}$). These findings suggest polluted air mixed with high concentrations of biogenic tracers induced more intense particulate formation and faster growth.

Additionally, on 23 March during the NPF2 event, concentrations of $H_2SO_4$ (below 0.2 10$^8$ molec.cm$^{-3}$) close to the ones observed on 27 March (anthropogenic NPF1 event type) were observed. Characterized as an apple event type, the particulate formation and growth rates were not calculated for the NPF1 event on 27 March and were hence not compared with the ones associated to NPF event on 23 March. Note that, higher strength (i. e. maximum particle cluster number concentration) was noticed for the selected NPF2 event day (maxima $N_{\text{PSM-DMPS}}$ of 63000 cm$^{-3}$ on 23 March and of 41000 cm$^{-3}$ on 27 March). Moreover, at higher $H_2SO_4$ concentrations than ones observed on 23 March, NPF1 event days have shown contrasted particulate formation rates ($J_{1.5}$: 5.00-49.60 cm$^{-3}$s$^{-1}$ on 18-25 March, respectively) and both higher mean grow rate and mean CS ($GR_{1.5-3}$: 4.75 nm.h$^{-1}$ – CS: 0.12 s$^{-1}$) than the mean ones associated to NPF2 event days ($GR_{1.5-3}$: 3.70 nm.h$^{-1}$ – CS: 0.09 s$^{-1}$). It seems that polluted air masses, observed at the receptor site during NPF1 event days, were characterized by a high amount of condensable species involved in the particle growth permitting to overcome the increased CS and allowing also a fast growth. At higher $H_2SO_4$ and BVOC concentrations, NPF2 event days occurring on 8-10 March have shown higher particulate formation rates than the one of NPF event of 23 March ($J_{1.5}$: 12.23 cm$^{-3}$s$^{-1}$ in average ± 5.62 cm$^{-3}$s$^{-1}$ on 8-10 March and $J_{1.5}$: 8.97 cm$^{-3}$s$^{-1}$ on 20 March). These findings again, would confirm polluted air mixed with high concentrations of anthropogenic tracers can induce more intense particulate formation.

As a result of these comparisons, chemical or photochemical reactions involving biogenic and anthropogenic species form new compounds may be involved in nucleation. Several field studies have found an enhancement of biogenic SOA under the influence of anthropogenic emissions (e. g. Carlton et al., 2010; Shilling et al., 2013). The enhancement can be a result of increased gas to particle partitioning, increased oxidant concentrations or a change in the reaction pathways (Hoyle et al., 2011; Kanakidou et al., 2000). Laboratory experiments have also shown higher SOA formation levels in mixtures of VOCs compared to single VOC (Ahlberg et al., 2017; Flores et al., 2014). Ahlberg et al., 2017 even found that



isoprene did not produce much SOA mass in single VOC experiments but contributed to the mass in the cases of VOC mixtures.

As a summary, NPF can occur at a various condensational sink and both under polluted and clean atmospheric conditions. Some NPF events can occur at CAO at low $SO_2$ concentrations and low CS under clean atmospheric conditions.

High calculated $H_2SO_4$ concentrations coupled with high BVOC concentrations seem to be one of the most favorable conditions to observe NPF at CAO in March 2015. Relatively high particulate formation and growth rates were associated to NPF event days of mixed origins suggesting an intense particulate formation and a fast growth. Higher strength was noticed for NPF2 event day under mixed influence (anthropogenic and biogenic – 23 March) than the ones observed both during NPF1 and NPF4 event days under anthropogenic and biogenic origins respectively, for the same levels of precursors

(anthropogenic and biogenic, respectively) suggesting combination of biogenic and anthropogenic species form new compounds which may be involved in nucleation. The next part of this section is focused on 8-10 March (mixed NPF event type) to better understand how the interaction of BVOC species with anthropogenic compounds can initiate nucleation and contribute to early growth of nucleated particles.

### 3.4.2 Focus on BVOC contributions to particle formation and growth: 8 - 10 March NPF events

Firstly, we will focus our discussion on the behavior of selected parameters during NPF events observed between 8-10 March (represented by yellow periods in Fig. 12), corresponding to the NPF2 type events. The three successive NPF events were all initiated at 08:00 LT occurring around 2 hours after sunrise when isoprene concentrations started to increase in agreement with temperature and sun radiation. Daytime variation (06:00-17:00 LT) of $N_{PSM-DMPS}$ was consistent with MVK+MACR one which could represent the oxidized species producing new particles by nucleation. Maximal $N_{PSM-DMPS}$

(between 30000 cm$^{-3}$ and 40000 cm$^{-3}$) were observed at 11:00-12:00 in agreement with daily maximal solar radiation and isoprene concentrations (between 100 and 160 ppt). However, daily maximal $N_{PSM-DMPS}$ decreased from day to day while slightly higher maximal isoprene daily concentrations were noticed on 9 and 10 March compared to 8 March. This finding might be linked with higher CS recorded at the beginning of NPF events on 9 and 10 March compared to 8 March, which could reduce formation of new particles. As stated before, $SO_2$ and $H_2SO_4$ concentrations in the 8-10 March period were

among the highest ones observed during the campaign (Fig. 10-11). $H_2SO_4$ and $SO_2$ concentrations started to increase at 09:00 LT on 8 March and at 10:00 LT on 9 March which also correspond to the increase of $N_{PSM-DMPS}$ concentrations (Fig. 12) suggesting that $H_2SO_4$ also play a role in NPF. Additionally, daily maximal $H_2SO_4$ and $SO_2$ concentration also decrease from 8 March to 10 March as particle cluster concentrations did. Monoterpene concentrations remained low during these NPF events (below 400 ppt) but higher concentrations were observed at night suggesting there possible contribution in the

growth of newly formed particles. Contrary, monoterpene oxidation products were shown to produce new particles by nucleation more efficiently than the isoprene oxidation products (Bonn et al., 2014; Spracklen et al., 2008). This finding could suggest that isoprene only may not contribute to particle nucleation while isoprene combined with anthropogenic





species can be involved in nucleation. Additionally, oxidation products of monoterpenes, such as pinonaldehyde or nopinone, may nucleate and condense at an early stage of the new particle formation (Sellegri et al., 2005).

As a summary, during these specific days of the campaign, nucleation mode particles may be formed by the combination of high $H_2SO_4$ and isoprene oxidation product concentrations under favorable meteorological conditions (high temperature and solar radiation and low relative humidity – Fig. 5), resulting in an increase of SV-/LV-OOA contributions. A similar trend was observed between CS and isoprene concentrations during NPF events only suggesting that these compounds do not only contribute to the nucleation mode for particles but also to aerosol growth directly after nucleation.

The focus is now shifted to the variability of selected parameters during nighttime succeeding these NPF events (periods represented by blue color on Fig. 12) partly to highlight the role of monoterpenes. These nighttime periods were characterized by relatively high CS (up to 0.16 s$^{-1}$), nocturnal temperatures among the highest ones observed during the campaign and relative humidity ranged between 50 % and 100 %. Note that, high isoprene concentrations have still observed even few hours after the sunset consistent with temperature variation (Sect. 3.1.2) and could also play a role during nighttime.

Blue periods on Fig. 12 highlight periods characterized by a clear increase of SV-OOA contributions while inorganic aerosols concentrations and LV-OOA contributions remained stable. The increase of SV-OOA occurred at high isoprene concentrations, sometimes at high monoterpene concentrations and at high CS, in favor of BVOC condensation onto pre-existing particles. Moreover, after 19:00 on 10 March, the highest monoterpene concentrations observed during the campaign (up to 4 ppb) coincided with an elevation of SV-OOA contributions up to 7.3 µg.m$^{-3}$, consistent with the fact that oxidation products of monoterpenes are known to contribute to particle growth (Birmili et al., 2003).

As a summary, BVOCs observed at night at CAO potentially play a role in particle growth by condensing onto pre-newly formed aerosols and significantly influence levels and variations mainly of SV-OOA. The relationship between BVOCs and OA stated in Debevec et al. (2017) was hence confirmed highlighting the importance of the local contribution.

## 4. Conclusions

The Eastern Mediterranean is considered as a sensitive region strongly impacted by air pollution making this location important to be investigated. This air pollution partly results of strong local anthropogenic emissions, particularly concentrated in coastal cities, natural emissions enhanced by favorable climatic conditions along with contributions of polluted air masses from 3 continents transported over long distances. All these combined sources of air pollutants will have impact on human health, ecosystems and climate. However, to clearly assess the various incidences of this complex pollution impacting the Mediterranean region, supplementary observational data collected in the region are needed since they remain scarce, especially in the Eastern Mediterranean. Given this background, an intensive field campaign was carried out during a 1-month period (March 2015) at a background Cypriot site within the framework of ChArMEx and ENVI-Med CyAr programs. In particular, this work focuses on the study of the sources and fates of BVOCs in the Eastern Mediterranean and based on the intense monitoring of isoprene, eight monoterpenes and seven OVOCs with on-line and off-line measurements.



Primary BVOCs were mainly composed of monoterpenes with peaks up to 4,500 ppt. α-Pinene and β-pinene were the major monoterpenes recorded (67 % of the total monoterpene concentration). Additionally, isoprene and monoterpenes present two distinct kinds of diurnal evolution (daily and nighttime maximum, for isoprene and α-pinene, respectively) underlining two different kinds of emissions sources. The monoterpene nocturnal pattern was imputed to

nocturnal emissions from monoterpene storing plants from the understorey vegetation (pines forests). The observed isoprene pattern followed the usual diel profile which depended on environmental parameters (temperature and solar radiation). However, isoprene emissions seemed to be carried on few hours after sunset and may be released by the source responsible for most of monoterpene emissions (oak and pine forests). To determine the dominant emissions drivers for biogenic species, pinenes and isoprene temporal evolution were studied simultaneously with meteorological parameters. BVOC emissions

were controlled by ambient temperature, precipitation and relative humidity. Significant changes in monoterpene mixing ratios occurred during and after rainy periods. Rainfall appeared to induce a stress factor onto the vegetation and therefore may have caused short-term increases in the release of monoterpenes from the vegetation that may stimulate diurnal sources of monoterpenes. High relative humidity and high temperature were favorable conditions to observe at the station high BVOC concentrations originating from the nocturnal biogenic sources.

BVOCs are known to have their importance in the growth and possibly also in the early stages of formation of atmospheric aerosol particles. Based on observations of the particle size distribution performed with a DMPS and the total number concentrations of particles larger than 1 nm diameter measured by PSM, NPF events were found to occur on 14 out of 20 days. For all suspected parameters having a role in NPF (meteorological parameters, $H_2SO_4$ and gaseous compounds), we present mean levels and daily variations during different classes of nucleation events and non-event days. NPF can occur

at a various CS and both under polluted and clean atmospheric conditions. High calculated $H_2SO_4$ concentrations coupled with high BVOC concentrations seem to be one of the most favorable conditions to observe NPF at CAO in March 2015. Relatively high particulate formation and growth rates were associated to NPF event days of mixed origins suggesting an intense particulate formation and a fast growth. Higher strength was also noticed for NPF event days of mixed origin (anthropogenic and biogenic – 23 March) compared to the ones observed both for NPF events solely of anthropogenic origin

or biogenic origin, respectively, for the same levels of precursors (anthropogenic and biogenic, respectively). That suggests that the interaction of biogenic and anthropogenic species enhances the potential of nucleation. A focus on specific NPF period (mixed event type) highlighted BVOC combination with anthropogenic compounds influenced nucleation formation and growth of newly particles. During these days, nucleation mode may be induced by the combination of high $H_2SO_4$ and isoprene concentrations and under favorable meteorological conditions (high temperature and solar radiation and low relative

humidity) and low CS, resulting in an increase of $SO_4$ concentrations but also an increase of SV-/LV-OOA contributions. BVOCs contributed as well to the aerosol growth by condensing onto pre-existing aerosols since high BVOC concentrations were observed during succeeding night of NPF events consistent with CS variations leading to a significant increase of SV-OOA contributions.




The list of BVOCS measured within this work is not exhaustive, future prospects should focus especially on the measurements of sesquiterpenes which are very reactive and of interest for NPF study.

**Acknowledgements**

This study was supported by ChArMEx, ENVI-MED and ACTRIS-2 (European Union's Horizon 2020 research and innovation programme, grant agreement No 654109), CEA and CNRS. Atmospheric observations performed in Cyprus have been partly supported by the EU-H2020 ACTRIS-2 project (European Union's Horizon 2020 research and innovation programme, grant agreement No 654109) and the EU FP7-ENV-2013 BACCHUS project (grant Agreement 603445). The authors would like to thank N. Mihalopoulos for his help in the establishment of the CAO observatory; F. Dulac and E. Hamonou for managing with enthusiasm the ChArMEx project. The present work is a contribution to the Labex CaPPA (Chemical and Physical Properties of the Atmosphere) funded by the French National Research Agency (ANR-11-LABX-005-01) and the European Funds for Regional Economic Development (FEDER). The "Hauts-de-France" Regional Council, the French Ministry for Higher Education and Research and the FEDER are also acknowledged for their financial support through the CPER research project CLIMIBIO (changement Climatique, dynamique de l'atmosphère, impacts sur la biodiversité et la santé humaine).



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




**Table 1: Technical details of the set-up for VOC measurement during the intensive field campaign from 1st March 2015 to the 29th of March 2015.**

[a] **ethane, ethylene, propene, propane, i-butane, n-butane, acetylene, i-pentane and n-pentane**

[b] **2-methylpentane, benzene, toluene, ethylbenzene, m,p-xylenes, o-xylene, α-pinene and β-pinene**

5  [c] **m33 (methanol), m42 (acetonitrile), m45 (acetaldehyde), m59 (acetone), m69 (isoprene), m71 (MVK+MACR and eventually Isoprene Hydroxy Hydroperoxide (ISOPOOH)), m73 (MEK), m79 (benzene), m93 (toluene), m107 (xylenes + $C_7$-species) and m137 (monoterpenes)**

| Instrument | On-line measures | | | Off-line measures | |
|---|---|---|---|---|---|
| | GC-FID ChromaTrap | GC-FID AirmoVOC | PTR-QMS Scan mode (33 amu – 137 amu) | DNPH cartridges – Chemical desorption (acetonitrile) – HPLC-UV | Solid adsorbent – Adsorption/thermal desorption – GC-FID |
| **Time Resolution (min)** | 30 | 30 | 10 | 180 | 180 |
| **Number of samples** | 1282 | 1321 | 3879 | 207 | 211 |
| **Temporal coverage (%)** | 94 | 97 | 93 | 88 | 90 |
| **Detection limit (ppt)** | 8-104 | 7-17 | 11-203 | 6-27 | <5 |
| **Uncertainties $\frac{U(X)}{X}$ (%) mean [min – max] (%)** | 39 [14-73] | 36 [18-53] | 22 [18-44] | [11-37] | [3-26] |
| **Calibrated species** | 9 $C_2$ - $C_5$ [a] VOCs | 9 $C_6$ - $C_{10}$ [b] VOCs | 11 mass [c] | 10 $C_1$ - $C_6$ OVOCs | 6 $C_6$ - $C_{11}$ n-aldehydes 15 $C_5$ - $C_{16}$ alkanes 9 $C_6$ - $C_9$ aromatics 9 Monoterpenes |
| **Reference** | Gros et al., 2011 | Xiang et al., 2012 | Blake et al., 2009; de Gouw and Warneke, 2007; Taipale et al., 2008 | Detournay, 2011; Detournay et al., 2013 | Detournay, 2011; Detournay et al., 2011; Ait-Helal et al., 2014 |





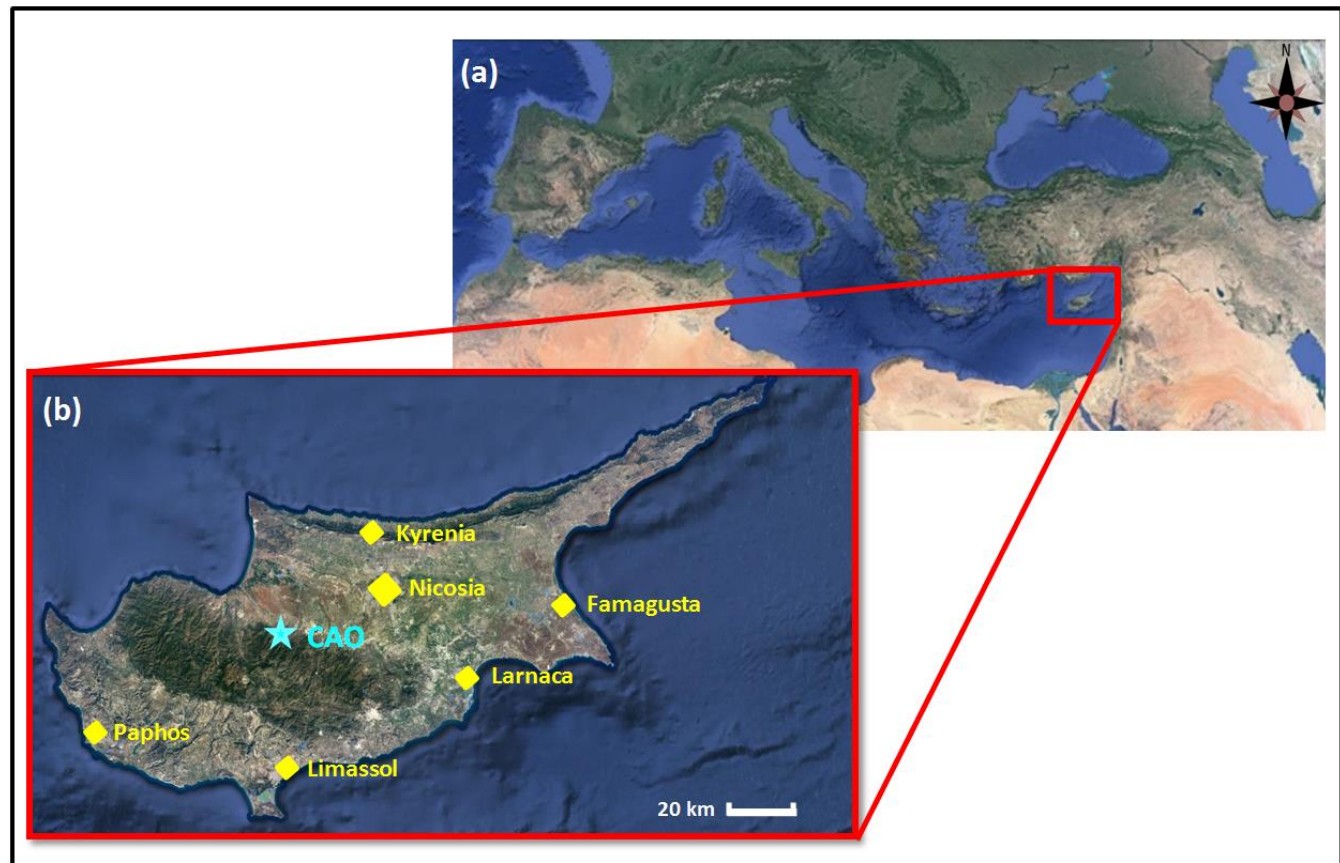

**Figure 1: Maps of Cyprus and Mediterranean region. (a) - Position of Cyprus in the Mediterranean region. (b) – The sampling site and major Cypriot agglomerations are displayed as blue star and yellow diamonds, respectively.**



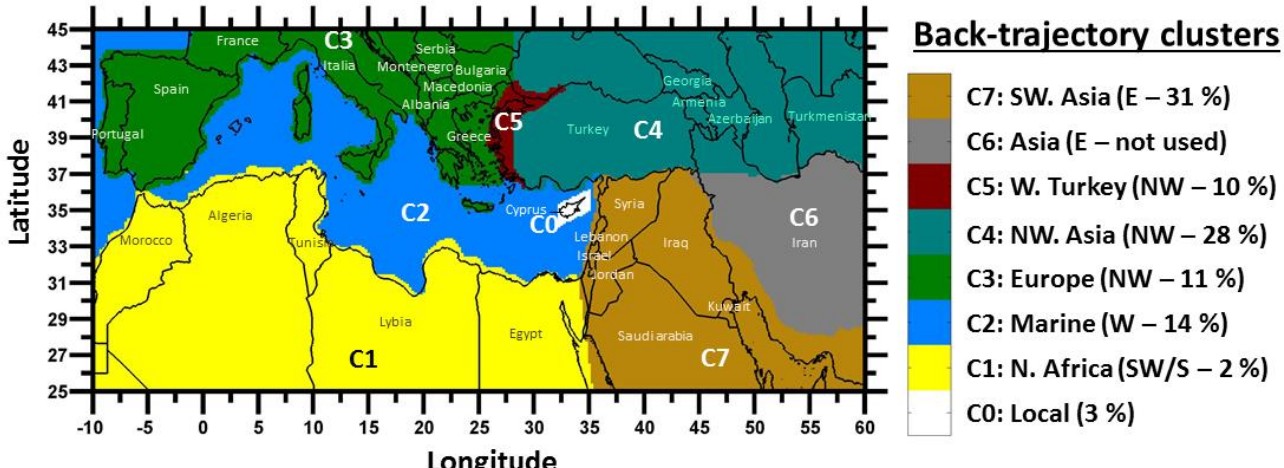

**Figure 2: Classification of air masses which impacted the site during the intensive field campaign of March 2015 and their relative contribution. A fraction of 2 % (not shown here) is attributed to air masses of mixed origins.**





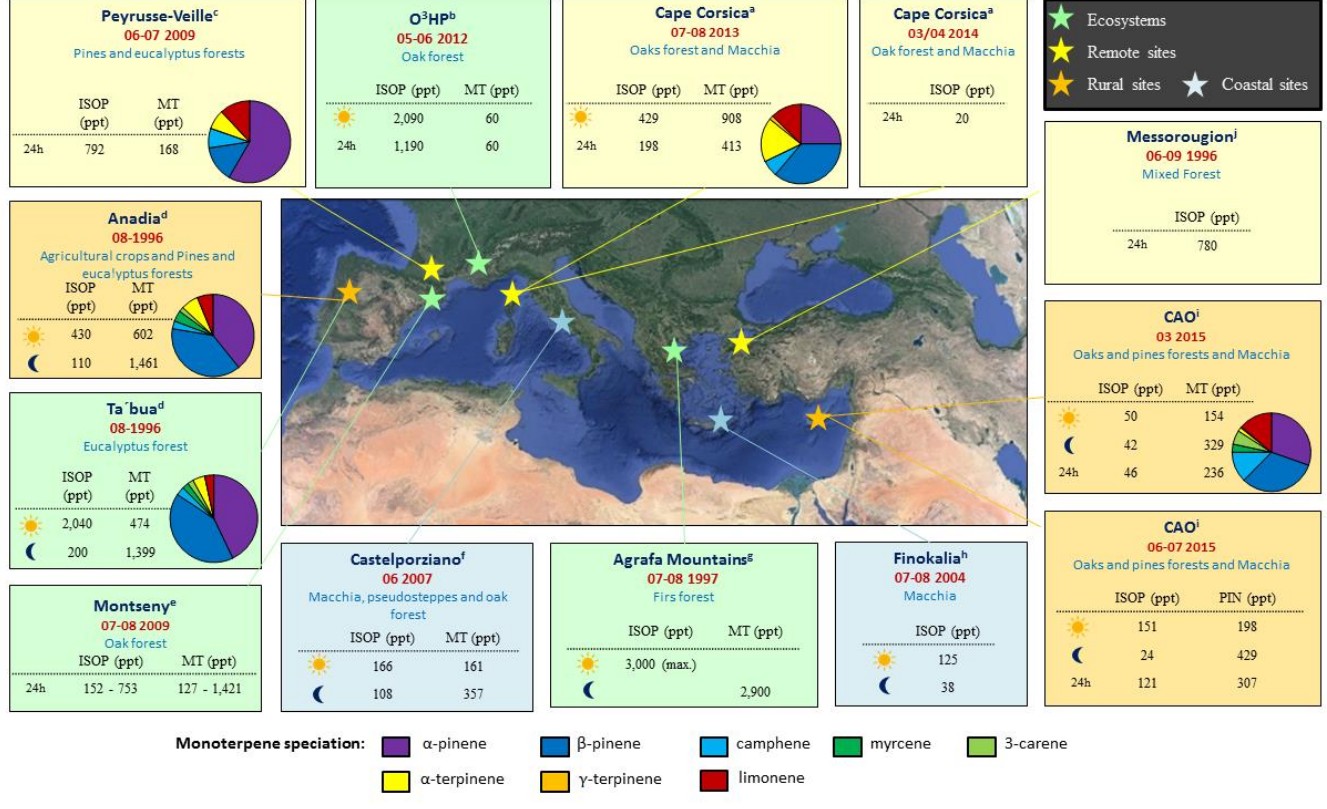

**Figure 3: Comparison of mean concentrations (in ppt) and speciation (in %) of primary BVOCs with the ones observed in the literature in the Mediterranean region with different vegetation types. "ISOP", "MT" and "PIN" are abbreviations respectively referring to isoprene, monoterpenes and pinenes.**

5    [a] Michoud et al., 2017; Kalogridis, 2014, ChArMEx database, [b] Kalogridis et al., 2014, [c] Detournay et al., 2013, [d] Cerqueira et al., 2003, [e] Seco et al., 2011, [f] Davison et al., 2009, [g] Harrison et al., 2001, [h] Liakakou et al., 2007, [i] this study, [j] Moschonas and Glavas, 2000.





**Figure 4: Diel variation of isoprene and monoterpenes, represented by hourly box plots (in green colors) in comparison with mean diel variation of meteorological parameters (solar radiation, temperature displayed as red lines and orange boxes, respectively). White marker represents the mean value, blue solid line represents the median value and the green box shows the InterQuartile Range (IQR). The bottom and the top of box depict the first and the third quartiles (i.e. Q1 and Q3). The ends of the whiskers correspond to the first and the ninth deciles (i.e. D1 and D9). Time is given in local time (UTC + 2 h).**









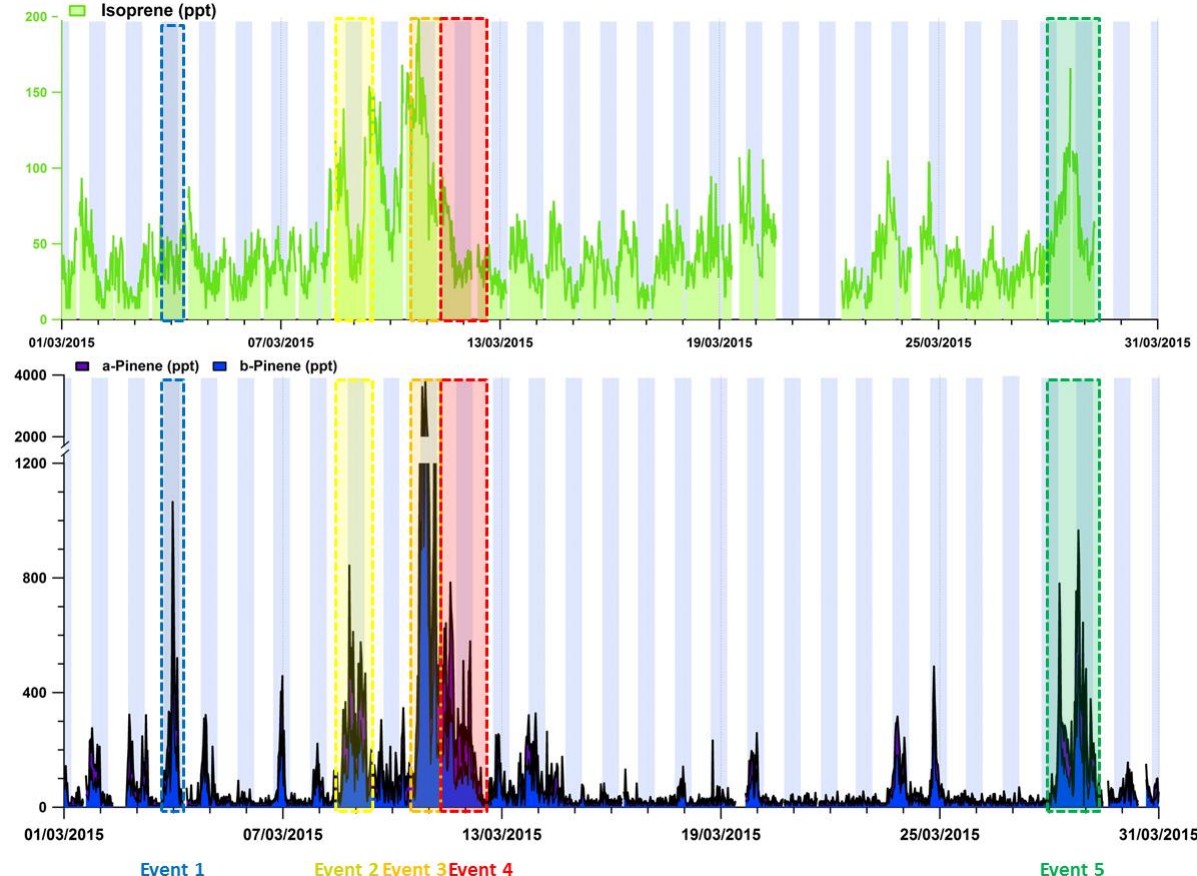

**Figure 5: Time series of isoprene and a selection of monoterpenes (α-pinene and β-pinene) in comparison with time series of meteorological parameters (solar radiation, temperature, wind speed, precipitation and relative humidity). Blue rectangles correspond to nighttime periods. BVOC events 1 to 5 referred to specific BVOC variations discussed in Sect. 3.2.**





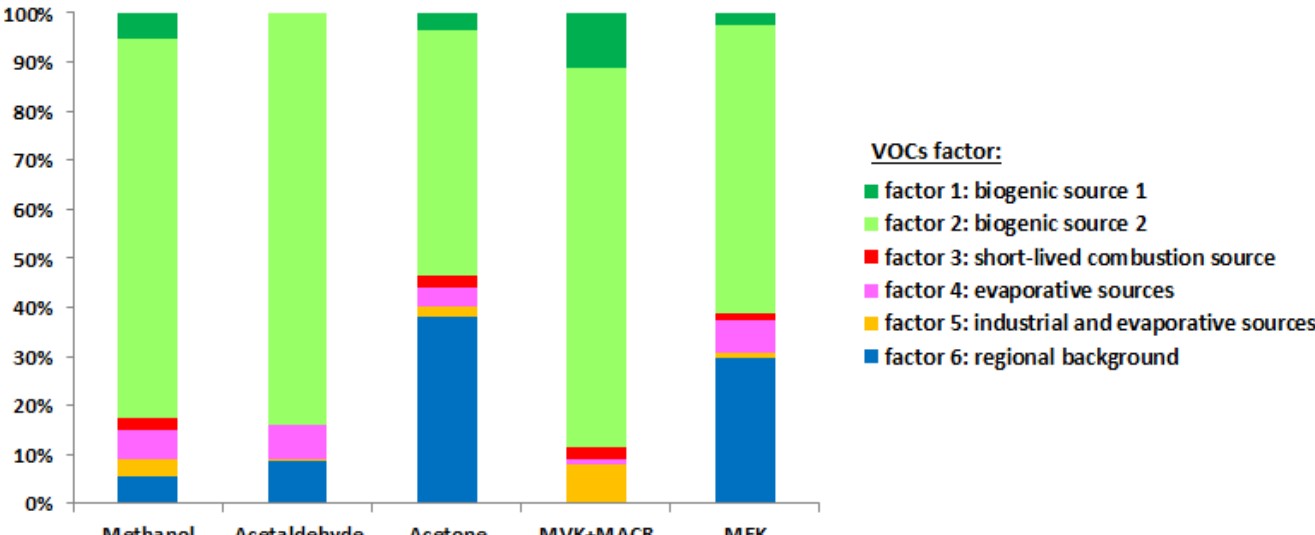

**Figure 6: PMF factor contributions to the measured concentration of selected OVOCs. PMF analysis is presented in Debevec et al. (2017).**









**Figure 7: Diel variation of methanol and acetaldehyde, represented by hourly box plots (in blue colors) in comparison with mean diel variation of meteorological parameters (solar radiation, temperature displayed as red lines and orange boxes, respectively) and isoprene and its oxidation products (in green colors). White marker represents the mean value, blue solid line represents the median value and the green box shows the interquartile range. The bottom and the top of box depict the first and the third quartiles (i.e. Q1 and Q3). The ends of the whiskers correspond to the first and the ninth deciles (i.e. D1 and D9). Time is given in local time (UTC + 2 h).**





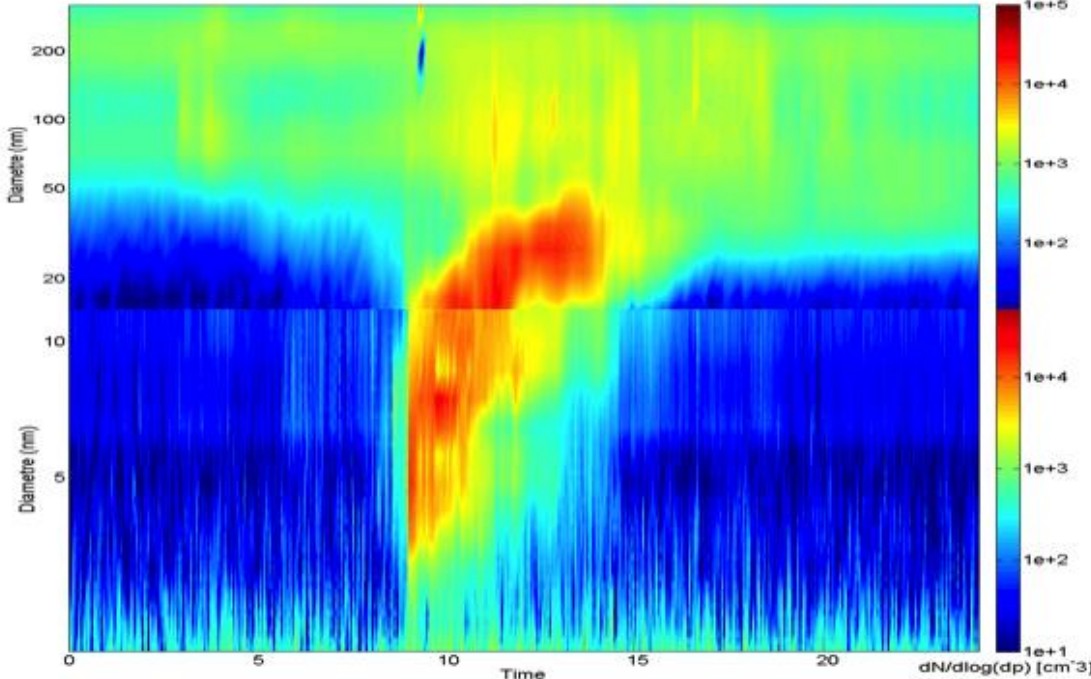

**Figure 8: Example of size distribution spectra, measured with DMPS and PSM, showing an NPF event of type Ia occurring on 14 March 2015 at the CAO station**









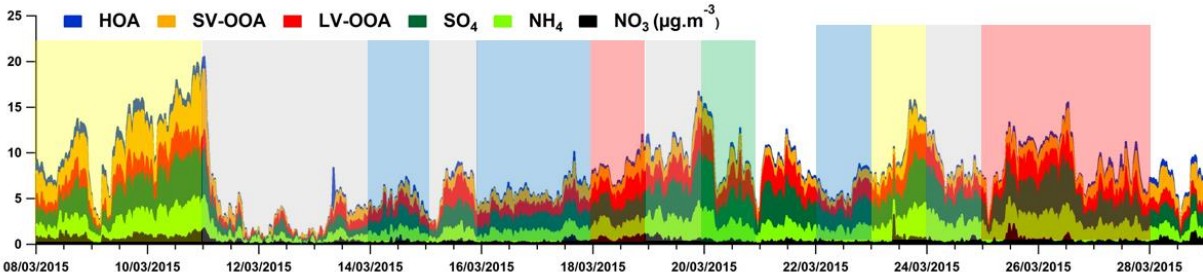

**Figure 9: Time series of particle number $N_{PSM}$, $N_{DMPS}$ and CS in comparison with suspected parameters controlling NPF events ($SO_2$, $H_2SO_4$, isoprene and monoterpenes) and accumulated time series of $PM_1$ contribution. The color code highlights NPF event days and non-event days (grey periods). Red periods represent NPF1 event days with anthropogenic origin. Yellow periods represent NPF2 event days both with mixed origins (anthropogenic and biogenic). Blue and green periods are respectively for NPF events of marine (NPF3) and biogenic origin (NPF4). Organic aerosol (OA) factors: HOA - hydrogen-like OA; SV-OOA – semi-volatile oxygen-like OA; LV-OOA – low-volatile oxygen-like OA. Time is given in local time (UTC + 2 h).**









**Figure 10: Average and standard deviation of CS, particles formation and growth rates ($J_{1.5}$ and $GR_{1.5-3}$, respectively) and atmospheric parameters daily concentrations measured at the CAO station in case of event (NPF1-NPF4) or non-event days (grey bars). Red bars represent NPF1 event days with anthropogenic origin. Yellow bars represent NPF2 event days both with mixed origins (anthropogenic and biogenic). Blue and green bars are respectively for NPF events of marine (NPF3) and biogenic origin (NPF4).**





**Figure 11: Diel variation of meteorological parameters (global solar radiation, relative humidity and temperature), BVOCs (isoprene and monoterpenes) and H₂SO₄ during NPF event days (NPF1-NPF4 displayed as red, yellow, blue and green lines, respectively) and non-event days (grey lines). Diel variations are represented by daily mean values associated with standard deviation when several days were combined. Time is given in local time (UTC + 2 h).**









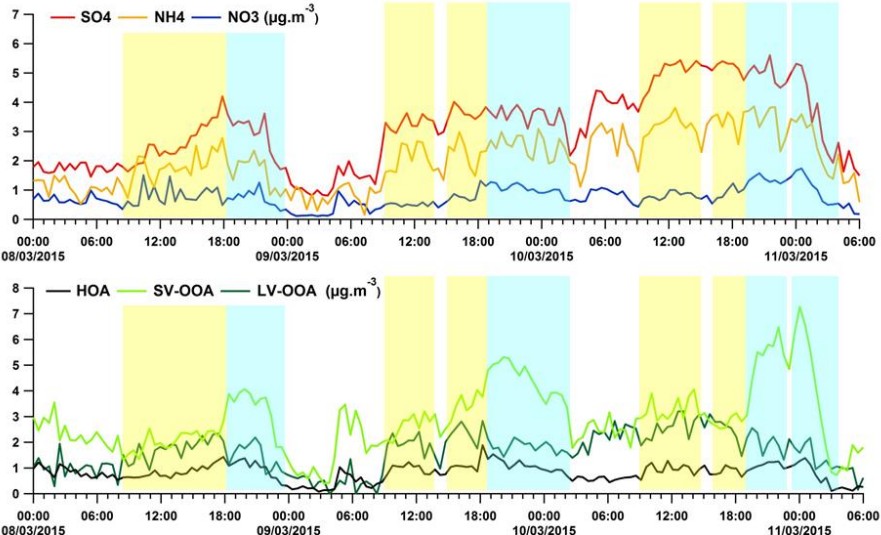

**Figure 12:** Time series of $N_{PSM}$, $N_{DMPS}$, $N_{PSM-DMPS}$ and CS during NPF2 event days (i. e. 08-10 March) in comparison with meteorological parameters (global solar radiation, temperature, relative humidity and precipitation), $SO_2$, $H_2SO_4$, BVOCs (isoprene, MVK+MACR and monoterpenes) and $PM_1$ composition. Time is given in local time (UTC + 2 h). NPF events are represented in yellow and nighttime succeeding these NPF events are depicted in blue. These periods are discussed in Sect. 3.4.2.