# Peer review of "Driving parameters of biogenic volatile organic compounds and consequences on new particle formation observed at an Eastern Mediterranean background site"

_Atmospheric Chemistry and Physics, 2018_

## Referee Comment (RC1) · Anonymous Referee #1 · 4 Jul 2018

The manuscript by Debevec et al. describe VOC mixing ratios measured at Cyprus Atmospheric Observatory during one month, March 2015. The motivation for the paper is to find out the driving factors for new particle formation. The measurements cover both on-line and off-line measurements, altogether more than 60 compounds were detected. These kind of intensive campaigns are valuable, since there is still a lot of unknown reactive organic compounds in the atmosphere. Unfortunately, the measurements did not cover sesquiterpenes, since they are likely to be very important in new particle formation and there is very little ambient data of sesquiterpene mixing ratios.

The paper is well written, it includes nice, informative figures and it suits well to be published ACP after minor revisions described below.

1. It is mentioned that ozone was removed in off-line sampling. MnO2 removes also sesquiterpenes, which is unfortunate, since SQTs are likely to have an important contribution in SOA formation. How about ozone removal in BVOC on-line measurements? How was it removed? In the VOC intercomparison between the used methods on-line measurements showed lower values than off-line measurements. Could this be due to different ozone removal efficiency?

2. It is not quite correct to say that monoterpenes were the most abundant group, when only monoterpenes, isoprene and few oxygenated compounds were measured.

3. Was isoprene measured with PTR-MS only? As mentioned, also other compounds than isoprene can add to m/z 69, and it would be interesting to see a comparison of isoprene measurement in supplement with other comparisons. High night-time isoprene concentrations could be due to other compounds/fragments too. Inomata et al. conducted such a comparison (ACP; doi:10.5194/acp-10-7085-2010) and found that isoprene measurements with PTR-MS were overestimated in comparison with FID.

4. It is not self-evident that MBO is temperature and light dependent in a same way as isoprene as mentioned. At least in boreal forest this could not be proved (Tarvainen et al., ACP doi.org/10.5194/acp-5-989-2005).

5. The chapter 3.2 has a misleading title. The manuscript deals with ambient mixing ratios, not with emissions. When discussing variability of the mixing ratios, the atmospheric mixing is not taken into consideration. There is a lot of discussion about the effect of humidity and rain in the mixing ratios, but these can be due to lower mixing layer height. There is currently also another paper under review in ACPD (Hellén et al., doi.org/10.5194/acp-2018-399) which claims that mixing layer height and temperature are the main factors determining ambient mixing ratios. Is there a way to evaluate mixing layer height at CAO if not measured? This would be extremely valuable and needs

to be taken into account.

6. Acetaldehyde is a known product of myriad of atmospheric reactions, for example OH radical reactions. OH radicals are produced only in sunlight and therefore acetaldehyde mixing ratios would also peak during daytime. The following sentence is from Millet et al. (ACP, 2010) abstract: "Hydrocarbon oxidation provides the largest acetaldehyde source in the model (128 Tg a-1, a factor of 4 greater than the previous estimate), with alkanes, alkenes, and ethanol the main precursors. There is also a minor source from isoprene oxidation". Why are the authors convinced that VOC oxidation is not the cause for high midday acetaldehyde mixing ratios, but light dependent emissions?

7. Table showing the mean mixing ratios of individual compounds would be helpful.

---

## Referee Comment (RC2) · Anonymous Referee #2 · 5 Jul 2018

The presented manuscript describes the online and off-line measurement of various organic compounds at a remote Mediterranean measurement site. The measurements include 20 days of data. The authors present characteristics of 4 different NPF classes, which they categorized based on air mass origin. The manuscript presents very interesting new results. I suggest minor revisions, described in the following.

1) My first comment is about the writing style of the manuscript. There are quite a lot of grammatical mistakes in the manuscript and it is very difficult to read. I suggest to ask a native English speaker to correct the language before re-submitting.

[Figure]

2) Why are you not showing any data from the NAIS measurements? It would be very interesting to see mean diel cycles for different size classes below 20 nm from the NAIS measurements for different NPF event day classes and non-event days. A comparison to PSM size classes and DMPS would be helpful in the same Figures.

3) A table, summarizing the findings regarding NPF event days and non-event days is needed. That table could contain the information that is shown in Figure 10 and 11, for the different NPF classes found in your analysis.

4) The presented Figures are extensive and contain a lot of information. Please do not use yellow in your Figures, it is very hard to read the content of the Figures if there are yellow lines.

5) It is sometimes difficult to extract all the information in the Figures. I will make some detailed suggestions in the following.

6) In Figure 4, it is not clear to me, what exactly is presented here? Do those Figures include all measurement days, NPF event days only or non-event days only? Please do not use yellow.

7) Figure 5 is very difficult to read, there is yellow on yellow and an extensive amount of information. I suggest to make mean diel cycle Figures, summarizing the different NPF event day classes you observed, showing the same parameters as in each panel of the current Figure.

8) Figure 7 again, please avoid yellow. I do not really understand the difference between the first and the second panel, other than the second panel shows the same information as Panel 1, with added Methanol diel cycle. Maybe those two can be summarized in one panel? If there is a good reason to keep the first two panels separated, please explain it somewhere. I am not sure, which days are summarized here? Does that Figure include all measurement days? NPF event days, non-event days?

9) For Figure 9 I have a very similar comment as for Figure 5. It is easier to understand

the information if the different NPF event day classes are summarized as mean diel cycle Figures. Again, yellow on yellow.

10) I guess you chose the NPF2 in Figure 12, because of the high isoprene concentrations during that event class. I suggest again instead of showing time series of each day separately, to show mean diel cycle plots for the presented parameters comparing NPF2 event days and non-event days before or after NPF2 event days. Again, please avoid yellow on yellow.

---

## Author Comment (AC1) · 3 Aug 2018

**acp-2018-297: "Driving parameters of biogenic volatile organic compounds and consequences on new particle formation observed at an Eastern Mediterranean background site"**

The manuscript by Debevec et al. describes VOC mixing ratios measured at Cyprus Atmospheric Observatory during one month, March 2015. The motivation for the paper is to find out the driving factors for new particle formation. The measurements cover both on-line and off-line measurements, altogether more than 60 compounds were detected. These kind of intensive campaigns are valuable, since there is still a lot of unknown reactive organic compounds in the atmosphere. Unfortunately, the measurements did not cover sesquiterpenes, since they are likely to be very important in new particle formation and there is very little ambient data of sesquiterpene mixing ratios. The paper is well written, it includes nice, informative figures and it suits well to be published in ACP after minor revisions described below.

**Authors' Responses to Referee #1**

We would like to thank the Referee #1 for her/his general feedback and each of her/his useful comments/questions for improving the quality of this manuscript. All comments addressed by both referees have been taken into account in the revised version of the manuscript. In this respect, several figures were notably modified included in the supplementary. Please note that figures numbers are now different in this new version.

In the present document, authors' answers to the specific comments addressed by Referee #1 are mentioned in **blue**, while changes made to the revised manuscript are shown in **green**.

The comments on the manuscript are listed as follows:

**1/** It is mentioned that ozone was removed in off-line sampling. $MnO_2$ removes also sesquiterpenes, which is unfortunate, since SQTs are likely to have an important contribution in SOA formation. How about ozone removal in BVOC on-line measurements? How was it removed? In the VOC intercomparison between the used methods on-line measurements showed lower values than off-line measurements. Could this be due to different ozone removal efficiency?

The intensive field campaign was initially carried out at the CAO to provide insights of the origins and fates of VOCs and aerosols in the Eastern Mediterranean, focusing on an extensive high time resolution in-situ measurements performed at a representative receptor site. Instruments deployed during this month make possible the measurement of many tracer compounds from various sources that have been observed at similar rural or remote sites (e.g. Lanz et al., 2008; Leuchner et al., 2015; Michoud et al., 2017; Sauvage et al., 2009; Vlasenko et al., 2009). Considering it was the first intensive field campaign realized at CAO, the authors did not expect to observe such elevated BVOC mixing ratios, especially in March, even if a biogenic potential was noticed considering the site is rather close of oak and pine forests. As a consequence, a specific GC sensitive enough to monitor sesquiterpenes was not deployed in this intensive field campaign. Additionally, sesquiterpenes can be measure with a PTR-MS as they can fragment partly at m/z 205 (Kim et al., 2009). Unfortunately, the authors decided to limit the scan mode of the PTR-MS to m/z 137, in order to have a better time

resolution and since no significant levels were noticed at higher m/z (check realized in late February 2015). In a future study, and following the suggestion of referee #1, it could be interesting to investigate sesquiterpene role in NPF formation at CAO.

We didn't use any ozone scrubber for on-line measurements (GCs and PTR-MS). However, as recommended by Detournay et al. (2011), different ozone scrubbers were used during the sampling of off-line measurements presented in section 2.2.1 in order to prevent any ozonolysis of the measured compounds. A KI ozone scrubber was installed upstream of the sampling onto DNPH cartridges, while a $MnO_2$ ozone scrubber was used for the multi-sorbent cartridges.

In addition to their on-line measurements, α-pinene and acetaldehyde were also measured by off-line techniques. α-Pinene was collected by multi-sorbent cartridges, analyzed after by GC-FID, while acetaldehyde was sampled with DNPH cartridges and analyzed by HPLC-UV. α-Pinene and acetaldehyde were hence chosen to see the potential influence of ozone on on-line measurement by the cross-checking of the results during the field campaign. On-line versus off-line measurements of α-pinene and acetaldehyde concentrations displayed a quite good correlation ($r^2$: 0.69 and 0.81 for α-pinene and acetaldehyde, respectively) and a slope close to one for both compounds (1.08 and 1.16 for α-pinene and acetaldehyde, respectively). Regarding these results, we think that potential interferences of ozone caused on VOCs measurements with GC systems were limited in this study.

As remarked by referee #1, on-line measurements of the sum of monoterpenes showed lower values than off-line measurements. The authors think that it could be partly due to β-pinene off-line measurements. The authors met some technical difficulties to correctly quantify β-pinene for the off-line method since the calibration results were not reproducible. It could notably affect the concentrations of the sum of monoterpenes for the off-line method, since β-pinene concentrations represented 32 % of the total monoterpene (i.e. the sum of 8 monoterpenes) concentrations.

Additionally, the authors consider that PTR-MS monoterpene measurements were more reliable than off-line monoterpene measurements considering their uncertainties (22 %) that's why on-line measurements were used in this study for the variability investigation of monoterpene concentrations.

**2/** It is not quite correct to say that monoterpenes were the most abundant group, when only monoterpenes, isoprene and few oxygenated compounds were measured.

Correction applied in the revised manuscript (Page 11, lines 1 – 2): *"Among BVOCs monitored during the intensive field campaign, the most abundant were monoterpenes."*

**3/** Was isoprene measured with PTR-MS only? As mentioned, also other compounds than isoprene can add to m/z 69, and it would be interesting to see a comparison of isoprene measurement in supplement with other comparisons. High nighttime isoprene concentrations could be due to other compounds fragments too. Inomata et al. conducted such a comparison (ACP; doi:10.5194/acp-10-7085-2010) and found that isoprene measurements with PTR-MS were overestimated in comparison with FID.

During the intensive field campaign, isoprene GC measurements were invalidated since we met technical problems (instability of the baseline around isoprene time response) which did not

make possible a correct identification and quantification for isoprene as such levels. This problem was solved after the intensive field campaign, by the replacement of the $H_2$ generator by compressed $H_2$ in bottle which permitted to reduced baseline interferences. As a consequence, isoprene was monitored with a GC-FID from late April 2015 to October 2015 and hence comparison of isoprene PTR-MS and GC measurements cannot be done. The authors agree with referee #1 on the fact that nighttime isoprene concentrations could be due to other compounds fragments which was specified in Section 2.2.1 of the revised manuscript.

Correction applied in the revised manuscript (page 6, lines 2-4):
"Note that, nighttime isoprene concentrations discussed in Sect. 3.1 and 3.2 could be due to other compound fragments such as 2-methyl-3-butene-2-ol (MBO)."

**4/** It is not self-evident that MBO is temperature and light dependent in a same way as isoprene as mentioned. At least in boreal forest this could not be proved (Tarvainen et al., ACP doi.org/10.5194/acp-5-989-2005).

It was nuanced in the revised version of the manuscript as fallowed (Page 13, lines 4-7):
" The emissions of MBO could require light as isoprene (Harley et al., 1998) or could be mainly temperature dependent (Hellén et al., 2018; Tarvainen et al., 2005). Isoprene and/or MBO that were emitted during the late afternoon could be not fully oxidized photochemically, as OH concentrations begin to fall, and could remain in the nighttime atmosphere."

**5/** The chapter 3.2 has a misleading title. The manuscript deals with ambient mixing ratios, not with emissions. When discussing variability of the mixing ratios, the atmospheric mixing is not taken into consideration. There is a lot of discussion about the effect of humidity and rain in the mixing ratios, but these can be due to lower mixing layer height. There is currently also another paper under review in ACPD (Hellén et al., doi.org/10.5194/acp-2018-399) which claims that mixing layer height and temperature are the main factors determining ambient mixing ratios. Is there a way to evaluate mixing layer height at CAO if not measured? This would be extremely valuable and needs to be taken into account.

Correction applied in the revised manuscript (page 13, lines 8-13):
"3.2 Factors controlling BVOC concentrations

In this section, time variations of main monoterpenes and isoprene are examined along with meteorological parameters in order to determine the dominant factors controlling BVOC concentrations."

As unfortunately the Planet Boundary Layer (PBL) height was not measured at CAO, we used instead PBL assimilated data conducted for the Troodos station (32.88° E - 34.92° N, ~20 km westerly from the CAO station) which are described in the section S1 added in the Supplement. The comparison of BVOC time series with PBL height was added to the section 3.2 of the revised manuscript and consequently, Figure 5 was modified in the revised manuscript.

Correction applied in the revised manuscript (page 8, lines 19-28):
"2.2.4 Meteorological measurements and assimilated data

Meteorological parameters (temperature, pressure, relative humidity, wind speed, wind direction and radiation) were monitored every 5 min using a weather station (Campbell Scientific Europe, Antony, France) located on the roof top of the CAO building, at approximately 5 m a.g.l. Additionally, Planet Boundary Layer (PBL) assimilated data were generated by the European Centre for Medium-Range Weather Forecast (ECMWF) Interim Re-Analysis (Era-Interim) global atmospheric reanalysis at the location corresponding to the Troodos station (32.88° E – 34.92° N, ~20 km westerly from the CAO station). ERA-Interim model, set-up and dataset are detailed in Sect. S1 in the Supplement. Even if these PBL data assimilated were not provided for the CAO station but for the Troodos one, they were only used in this study to qualitatively investigate PBL height effect on BVOC concentration levels and variations."

Additional information added in the revised manuscript (from page 13, line 14 to page 14, line 5):

[revised manuscript text omitted]

The ERA-Interim dataset starts from 1979 and continues to provide information until present in near real-time. Gridded data products include a large variety of 3-hourly surface parameters, describing weather as well as ocean-wave and land-surface conditions, and 6-hourly upper-air parameters covering the troposphere and stratosphere. Vertical integrals of atmospheric fluxes, monthly averages for many of the parameters, and other derived fields have also been produced. Berrishford et al. (2011) provide a detailed description of the ERA-Interim product archive. ERA-Interim products are normally updated once per month, with a delay of two months to allow for quality assurance and for correcting technical problems. The ERA-Interim atmospheric model has a spatial resolution of 0.75°x0.75° and expands vertically with 60 atmospheric layers. The reanalysis product is produced with a sequential data assimilation scheme, using 12-hourly analysis cycles , a time-window when available observations are assimilated into the information from the forecast model as described in Dee et al. (2011).

The ERA-Interim model includes a PBL height parameter calculated from the Bulk Richardson number (Troen and Mahrt, 1986), which is based on ratios of both dynamic and thermodynamic vertical gradients and hence characterizes the degree of turbulence. Given the fact that the boundary layer is often associated with stronger mixing (as compared to the free troposphere) due to increased levels of turbulence, it would be natural to investigate properties associated with turbulence. Essentially, the PBL height is defined as the level where the bulk Richardson number reaches a critical value of 0.25, based on the difference between quantities at this level and the lowest model level as an estimator for the vertical stability. Bulk Richardson number is available a 6-h and 12-h forecasts.

However, as reported in von Engeln and Teixeira (2013) this method of estimating the stability from dry thermodynamic variables (not moist), tends to provide estimates of PBL height that is often closer to the cloud-base height in marine cloudy boundary layers, rather than the PBL height itself (Janssen and Bidlot, 2003). Seidel et al. (2012) reports that for their scope of assessing the climatology of the planetary boundary layer over the continental United States and Europe with the use of ERA-Interim datasets, they did not employ the estimates of the BLH from ERA-Interim itself, because they are computed using an algorithm not applicable to radiosonde data (due to the fact that turbulence parameters are required for this application). With a preliminary analysis, they report that the ERA-interim PBL height product (i.e., with the ECMWF algorithm) shows higher heights, especially over high elevation regions, than the algorithm used in their study on the radiosonde data. Differences were below 100 m at night and of several 100 m during daytime."

Revised Figure 5 is the following:

[Figure]

[Figure]

**Figure 5: Time series of isoprene and a selection of monoterpenes (α-pinene and β-pinene) in comparison with time series of meteorological parameters (boundary layer height, wind speed, solar radiation, temperature, precipitation and relative humidity). Blue rectangles correspond to nighttime periods. BVOC episodes 1 to 5 referred to specific BVOC variations discussed in Sect. 3.2. Note that, PBL assimilated data were generated by the ECMWF Era-Interim global atmospheric reanalysis at the location corresponding to the Troodos station (32.88° E – 34.92° N, ~20 km westerly from the CAO station).**

**6/** Acetaldehyde is a known product of myriad of atmospheric reactions, for example OH radical reactions. OH radicals are produced only in sunlight and therefore acetaldehyde mixing ratios would also peak during daytime. The following sentence is from Millet et al. (ACP, 2010) abstract: "Hydrocarbon oxidation provides the largest acetaldehyde source in the model (128 Tg a-1, a factor of 4 greater than the previous estimate), with alkanes, alkenes, and ethanol the main precursors. There is also a minor source from isoprene oxidation". Why are the authors convinced that VOC oxidation is not the cause for high midday acetaldehyde mixing ratios, but light dependent emissions?

From the 6 PMF factors reported in Debevec et al. (2017), the measured OVOCs were distributed among their different sources (Fig. 6). More than 80 % of the respective total mass of methanol, acetaldehyde and MVK+MACR was explained by biogenic sources, namely by factor 2 driven by isoprene emissions. The morning increase pattern of acetaldehyde concentration is similar to isoprene pattern rather than isoprene oxidation product one that could suggest that acetaldehyde was mostly released in the atmosphere by local vegetation rather than produced by VOC oxidation processes in March 2015.

Correction applied in the revised manuscript (Page 16, lines 15-24):
"Hydrocarbon oxidation (mostly alkanes and alkenes but also isoprene and ethanol) provides the largest acetaldehyde source in the budget estimates of Millet et al. (2010). Nonetheless, for all reaction pathways of isoprene with atmospheric oxidants, acetaldehyde is produced as a second- or higher-generation oxidation product of isoprene (Millet et al., 2010). In addition to photochemical production, acetaldehyde is emitted by terrestrial plants, as a result of fermentation reactions leading to ethanol production in leaves and roots (Jardine et al., 2008; Rottenberger et al., 2008; Winters et al., 2009).  Acetaldehyde concentrations started to increase in the morning and peaked at midday followed by a gradual decrease throughout the rest of the day. The morning increase

pattern of acetaldehyde concentrations is similar to the isoprene pattern rather than isoprene oxidation product one and isoprene and acetaldehyde  correlated well ($r^2 = 0.49$). These findings  that acetaldehyde was mostly released in the atmosphere by local vegetation rather than produced by VOC oxidation processes in March 2015.

**7/** Table showing the mean mixing ratios of individual compounds would be helpful

The table showing statistics of mixing ratios of individual compounds was added to the Supplement of the revised manuscript as Table SI-1.

Table SI-1 in the Supplement is the following:

Table S1: Statistics ($\mu$g.m$^{-3}$), detection limits (DL - $\mu$g.m$^{-}$3) and relative uncertainties u(X)/X (Unc. - %) of selected VOC concentrations measured at the site.

| | Species | Min | 25 % | 50 % | Mean | 75 % | Max | σ | DL | Unc. |
|---|---|---|---|---|---|---|---|---|---|---|
| **DIENE** | **Isoprene** | 4 | 26 | 38 | 46 | 53 | 219 | 28 | 21 | 11 |
| **TERPENES** | **α-Pinene** | 8 | 8 | 18 | 58 | 58 | 1874 | 131 | 16 | 10 |
| | **β-Pinene** | 6 | 6 | 18 | 61 | 57 | 1962 | 142 | 12 | 12 |
| | **Camphene** | <1 | 5 | 11 | 25 | 29 | 275 | 37 | 1 | ND |
| | **Myrcene** | <1 | 2 | 4 | 6 | 8 | 43 | 7 | 2 | ND |
| | **Δ³-Carene** | <1 | 4 | 8 | 11 | 15 | 91 | 11 | 1 | ND |
| | **α-Terpinene** | <1 | 1 | 2 | 3 | 5 | 32 | 4 | 1 | ND |
| | **γ-Terpinene** | <1 | <1 | <1 | <1 | 1 | 12 | 2 | 1 | ND |
| | **Limonene** | <1 | 8 | 17 | 27 | 32 | 347 | 37 | 1 | ND |
| **ALCOHOL** | **Methanol** | 654 | 1658 | 2426 | 2765 | 3452 | 9074 | 1452 | 180 | 21 |
| **CARBONYL COMPOUNDS** | **Formaldehyde** | 399 | 678 | 909 | 986 | 1170 | 2416 | 409 | 25 | ND |
| | **Acetaldehyde** | 102 | 277 | 390 | 431 | 531 | 1533 | 209 | 44 | 10 |
| | **Acetone** | 423 | 861 | 1048 | 1083 | 1214 | 2662 | 335 | 17 | 9 |
| | **MVK+MACR** | 3 | 19 | 26 | 30 | 35 | 139 | 18 | 3 | 12 |
| | **MEK** | 59 | 154 | 196 | 210 | 242 | 653 | 84 | 13 | 9 |

ND: not determined

---

## Author Comment (AC2) · 3 Aug 2018

**acp-2018-297: "Driving parameters of biogenic volatile organic compounds and consequences on new particle formation observed at an Eastern Mediterranean background site"**

The presented manuscript describes the on-line and off-line measurements of various organic compounds at a remote Mediterranean measurement site. The measurements include 20 days of data. The authors present characteristics of 4 different NPF classes, which they categorized based on air mass origin. The manuscript presents very interesting new results. I suggest minor revisions, described in the following.

**Authors' Responses to Referee #2**

We would like to thank the Referee #2 for her/his general feedback and each of her/his useful comments/questions for improving the quality of this manuscript. All comments addressed by both referees have been taken into account in the revised version of the manuscript. In this respect, several figures were notably modified and included in the supplementary. Please note that figures numbers are now different in this new version.

In the present document, authors' answers to the specific comments addressed by Referee #2 are mentioned in **blue**, while changes made into the revised manuscript are shown in **green**.

The comments on the manuscript are listed as follows:

**1/** About the writing style of the manuscript, there are quite a lot of grammatical mistakes in the manuscript and it is very difficult to read. I suggest asking a native English speaker to correct the language before re-submitting.

The revised manuscript was corrected by a native English speaker. The referee #2 is invited to look at the peer review version of the revised manuscript in order to see all the modifications made consequently to his/her comment.

**2/** Why are you not showing any data from the NAIS measurements? It would be very interesting to see mean diel cycles for different size classes below 20 nm from the NAIS measurements for different NPF event day classes and non-event days. A comparison to PSM size classes and DMPS would be helpful in the same figures.

In this study, daily size distribution spectra measured with NAIS were mainly used to strengthen the identification and the classification of NPF events. In fact, the authors followed the classification scheme of Yli-Juuti et al. (2009), combining visual observation of NPF events from (N)AIS and DMPS measurements. The evolution of particle size distributions also gives us a way to know their growth and nucleation rates.

Additionally, during the intensive field campaign, the PSM was not operated in the scan mode (for the measurement of all particles having a diameter between 1 and 2.5 nm) but in the

total mode (for the measurement of all particles larger than 1 nm), which did not allow any growth and nucleation rate calculation in the size range 1-3 nm.

As mentioned by referee #2, there are already numerous Figures containing many information that's why the authors prefer not showing NAIS mean diel cycles in this study. Moreover, the aim of this study is not to provide an extensive investigation of NPF events but rather to focus on the role of BVOCs in the early stages of formation and the growth of atmospheric aerosol particles.

In the preliminary study, each DMPS size class was investigated individually. During NPF events, number concentrations of larger size particle classes can increase with a delay compared to number concentrations of first particle size class (20-27 nm from 8 to 11 March and 10-13 nm from 12 to 27 March). An example is provided in Figure X1, in agreement with the banana-shape depicted in Figure 8 of the manuscript.

Furthermore, to summarize the results and according to the aim of this study, the authors only made the distinction between $N_{PSM-DMPS}$, corresponding to number concentrations of sub-20 nm particles (from 8 to 11 March) or sub-10 nm particles (from 12 to 27 March) and $N_{DMPS}$, corresponding to number concentrations of either 20-200 nm particles (from 8 to 11 March) or 10-250 nm particles (from 12 to 27 March) in the manuscript. These two parameters can provide information on the early stages of formation (regarding $N_{PSM-DMPS}$) and the growth ($N_{DMPS}$) of atmospheric aerosol particles. As a result, $N_{PSM}$ was decomposed into $N_{DMPS}$ and $N_{PSM-DMPS}$ in Figure 9 of the revised manuscript (see authors' response 9). The authors hope that Figure 9 of the revised manuscript will meet referee #2 expectations about comparisons of PSM and DMPS measurements in the same Figures.

[Figure]

**Figure 8: Example of size distribution spectra, measured with DMPS and NAIS, showing an NPF event of type Ia occurring on 14 March 2015 at the CAO station**

[Figure]

**Figure X1: Time series of a selection of number concentrations, measured with the DMPS, showing an NPF event of type Ia occurring on 14 March 2015 at the CAO station**

**3/** A table, summarizing the findings regarding NPF event days and non-event days is needed. That table could contain the information that is shown in Figures 10 and 11, for the different NPF classes found in your analysis.

As proposed by referee #2, Figure 10 (of the initial version of the manuscript) was removed and converted into a table (as Table 2 in the revised manuscript) showing mean and standard deviation values for atmospheric parameters (supporting the classification of event days) along with property indicators for NPF events and factors with suspected influence on nucleation events.

Otherwise, Figure 11 was kept, and hence not integrated to Table 2, since the importance of diurnal variations as point out by referee #2 (please see authors' response 9 in complement). Nevertheless, mean and standard deviation values for some meteorological parameters (temperature, relative humidity and solar radiation) were added to Table 2 of the revised manuscript.

Table 2 of the revised manuscript is the following:

**Table 2: Average and standard deviation of CS, particle formation and growth rates ($J_{1.5}$ and $GR_{1.5-3}$, respectively), meteorological parameters (temperature, relative humidity and solar radiation) and atmospheric parameters daily concentrations measured at the CAO station in case of event (NPF1-NPF4) or non-event days.**

| Parameter | NPF1 event days | NPF2 event days | NPF3 event days | NPF4 event days | Non-event days |
|---|---|---|---|---|---|
| CS ($s^{-1}$) | $0.12 \pm 0.02$ | $0.09 \pm 0.02$ | $0.08 \pm 0.01$ | 0.12 | $0.07 \pm 0.04$ |
| $J_3$ ($cm^{-3}s^{-1}$) | 5.0 | $11.4 \pm 4.9$ | $6.4 \pm 1.4$ | 8.1 | - |
| $GR_{1.5-3}$ ($nm.h^{-1}$) | 5.0 | $3.7 \pm 1.6$ | $1.9 \pm 0.6$ | 2.8 | - |
| $PM_1$ ($\mu g.m^{-3}$) | $9.7 \pm 1.4$ | $12.9 \pm 2.8$ | $5.9 \pm 0.7$ | 9.8 | $6.4 \pm 3.6$ |
| $SO_4$ ($\mu g.m^{-3}$) | $2.9 \pm 0.7$ | $3.3 \pm 1.0$ | $1.9 \pm 0.3$ | 3.1 | $1.9 \pm 1.3$ |
| $NH_4$ ($\mu g.m^{-3}$) | $1.9 \pm 0.4$ | $2.1 \pm 0.6$ | $1.2 \pm 0.2$ | 1.8 | $1.2 \pm 0.8$ |
| $NO_3$ ($\mu g.m^{-3}$) | $0.5 \pm 0.2$ | $0.7 \pm 0.2$ | $0.3 \pm 0.1$ | 0.3 | $0.3 \pm 0.1$ |
| OM ($\mu g.m^{-3}$) | $4.3 \pm 0.4$ | $6.8 \pm 1.5$ | $2.6 \pm 0.3$ | 4.5 | $2.9 \pm 1.5$ |
| HOA ($\mu g.m^{-3}$) | $0.4 \pm 0.1$ | $0.7 \pm 0.2$ | $0.3 \pm 0.1$ | 0.4 | $0.3 \pm 0.1$ |
| SV-OOA ($\mu g.m^{-3}$) | $1.3 \pm 0.2$ | $2.5 \pm 0.9$ | $0.8 \pm 0.2$ | 1.1 | $0.9 \pm 0.4$ |
| LV-OOA ($\mu g.m^{-3}$) | $1.7 \pm 0.2$ | $1.8 \pm 0.5$ | $1.3 \pm 0.1$ | 2.2 | $1.3 \pm 0.7$ |
| BC ($\mu g.m^{-3}$) | $0.5 \pm 0.1$ | $1.0 \pm 0.3$ | $0.3 \pm 0.1$ | 0.4 | $0.3 \pm 0.1$ |
| CO (ppb) | $158.2 \pm 5.5$ | $162.5 \pm 9.2$ | $160.1 \pm 19.5$ | 155.1 | $151.6 \pm 13.2$ |
| $NO_2$ (ppb) | $1.1 \pm 0.2$ | $1.4 \pm 0.5$ | $0.8 \pm 0.1$ | 0.7 | $0.6 \pm 0.2$ |
| $SO_2$ (ppb) | $0.7 \pm 0.3$ | $0.7 \pm 0.3$ | $0.3 \pm 0.1$ | 0.2 | $0.2 \pm 0.1$ |
| $H_2SO_4$ ($molec.cm^{-3}$) | $6.3\ 10^7 \pm 5.2\ 10^7$ | $1.4\ 10^8 \pm 8.4\ 10^7$ | $4.3\ 10^7 \pm 1.8\ 10^7$ | $1.8\ 10^7$ | $2.3\ 10^7 \pm 1.7\ 10^7$ |
| Isoprene (ppt) | $34 \pm 7$ | $79 \pm 29$ | $33 \pm 7$ | 57 | $47 \pm 16$ |
| MVK+MACR (ppt) | $27 \pm 4$ | $61 \pm 23$ | $25 \pm 1$ | 26 | $30 \pm 8$ |
| Monoterpenes (ppt) | $115 \pm 19$ | $361 \pm 209$ | $148 \pm 80$ | 130 | $306 \pm 204$ |
| $O_3$ (ppb) | $50.4 \pm 3.7$ | $48.2 \pm 2.8$ | $46.4 \pm 2.6$ | 48.2 | $46.5 \pm 4.3$ |
| Temperature (°C) | $14.2 \pm 2.4$ | $15.4 \pm 3.7$ | $11.8 \pm 2.4$ | 10.7 | $11.2 \pm 1.7$ |
| Relative Humidity (%) | $54.0 \pm 12.3$ | $63.5 \pm 18.1$ | $61.3 \pm 9.6$ | 63.8 | $79.6 \pm 12.5$ |
| Solar radiation ($W.m^{-2}$) | $258 \pm 213$ | $255 \pm 192$ | $305 \pm 228$ | 283 | $203 \pm 199$ |

**4/** The presented Figures are extensive and contain a lot of information. Please do not use yellow in your Figures, it is very hard to read the content of the Figures if there are yellow lines.

An effort was realized to limit the use of yellow/light orange colors in the Figures of the manuscript. As a consequence, the color used to represent NPF event days categorized by a mixed (anthropogenic/biogenic) influence is now a dark orange (instead of yellow) in order to stay consistent with colors used for NPF event days of individual origin (i.e. red for NPF1 event days of anthropogenic origin and green for NPF3 event days of biogenic origin). The orange color used to represent solar radiation and $NH_4$ data has been darkened and $H_2SO_4$ concentrations are now represented in violet (instead of light orange).

The modifications applied to Figures 5, 9, 10 and 11 (of the initial version of manuscript) are explicit in the following answers.

**5/** It is sometimes difficult to extract all the information in the Figures. I will make some detailed suggestions in the following.

The authors thank referee #2 for these detailed suggestions which the authors will take into account in the following.

**6/** In Figure 4, it is not clear to me, what exactly is presented here? Do those Figures include all measurement days, NPF event days only or non-event days only? Please do not use yellow.

In Figure 4 is presented diurnal variations of isoprene and monoterpenes concentrations. These diurnal variations are also compared with mean diel variations of meteorological parameters (temperature and solar radiation) which are known to influence BVOC emissions, and so indirectly BVOC concentration variations.

This figure includes all BVOC measurement days with a PTR-MS, i.e. from 1 March to 29 March 2015, which has been specified in the caption of Figure 4. This period includes NPF event days and non-event days as the variation of BVOC concentrations was independent of this element.

As suggest by referee #2, the orange color used to represent solar radiation data has been darkened.

Revised Figure 4 is the following:

[Figure]

**Figure 4: Diel variation of isoprene and monoterpenes, represented by hourly box plots (in green colors) in comparison with mean diel variation of meteorological parameters (solar radiation, temperature displayed as red lines and orange boxes, respectively). This figure includes all BVOC measurement days with a PTR-MS (i.e. from 1 to 29 March 2015). White marker represents the mean value, blue solid line represents the median value and the green box shows the InterQuartile Range (IQR). The bottom and the top of box depict the first and the third quartiles (i.e.**

**Q1 and Q3). The ends of the whiskers correspond to the first and the ninth deciles (i.e. D1 and D9). Time is given in local time (UTC + 2 h).**

**7/** Figure 5 is very difficult to read, there is yellow on yellow and an extensive amount of information. I suggest making mean diel cycle Figures, summarizing the different NPF event day classes you observed, showing the same parameters as in each panel of the current Figure.

In Figure 5 is presented times variations of main monoterpenes and isoprene examined along with meteorological parameters in order to determine the dominant drivers for variations of BVOC concentrations.

Given the extensive amount of information for the Figure 5, the investigation of meteorological parameter effects on BVOC concentrations was mainly based on the study of 5 specific periods among the 29 days of BVOC measurements, called "events" in the initial version of manuscript. Otherwise, the appellation of "event" does not refer to NPF event day. Thanks to referee #2 comment, the authors realized that the use of the term "event" in this section can lead to confusion. As a result, the 5 specific periods are now called "episodes" in the revised manuscript.

The suggestion of referee #2 in making mean diel cycle Figures, summarizing the different NPF event day classes observed is relevant. The authors hope that Figure 10 of the revised manuscript (see authors' response 9) showing mean diel variations for some meteorological parameters (solar radiation, relative humidity and temperature) and BVOCs (isoprene and monoterpenes) among others meets referee #2's expectations on this point.

As suggested by referee #2, the orange color used to represent solar radiation data has been darkened.

Figure 5 of the revised manuscript is the following:

[Figure]

[Figure]

**Figure 5:** Time series of isoprene and a selection of monoterpenes (α-pinene and β-pinene) in comparison with time series of meteorological parameters (boundary layer height, wind speed, solar radiation, temperature, precipitation and relative humidity). Blue rectangles correspond to nighttime periods. BVOC episodes 1 to 5 referred to specific BVOC variations discussed in Sect. 3.2. Note that, PBL assimilated data were generated by the ECMWF Era-Interim global atmospheric reanalysis at the location corresponding to the Troodos station (32.88° E – 34.92° N, ~20 km westerly from the CAO station).

**8/** Figure 7 again, please avoid yellow. I do not really understand the difference between the first and the second panel, other than the second panel shows the same information as Panel 1, with added Methanol diel cycle. Maybe those two can be summarized in one panel? If there is a good reason to keep the first two panels separated, please explain it somewhere. I am not sure, which days are summarized here? Does that Figures include all measurement days? NPF event days, non-event days?

The first panel of Figure 7 highlights the delay of about 1 hour in the peak values between isoprene and its first oxidation products (MVK+MACR). On the second panel, BVOC concentrations are scaled differently than on the first one, which may make less obvious the occurrence of this delay. Considering the recommendation of referee #2, the panel 1 was moved to the Supplement (as Figure SI-3), in order to make Figure 7 of the revised manuscript less extensive.

Figure 7 includes all measurement days with a PTR-MS, i.e. from 1 March to 29 March 2015, which has been explicit in its caption. This period includes NPF event days and non-event days since the variation of acetaldehyde and methanol concentrations was studied independently from this element.

Figure 7 of the revised manuscript is the following:

[Figure]

**Figure 7: Diel variation of methanol and acetaldehyde, represented by hourly box plots (in blue colors) in comparison with mean diel variation of meteorological parameters (solar radiation, temperature displayed as red lines and orange boxes, respectively) and isoprene and its oxidation products (in green colors). This figure includes all measurement days with a PTR-MS (i.e. from 1 to 29 March 2015). White marker represents the mean value, blue solid line represents the median value and the green box shows the interquartile range. The bottom and the top of box depict the first and the third quartiles (i.e. Q1 and Q3). The ends of the whiskers correspond to the first and the ninth deciles (i.e. D1 and D9). Time is given in local time (UTC + 2 h).**

Figure SI-3 of the Supplement is the following:

[Figure]

**Figure SI-3: Mean diel variation of isoprene and its oxidation products (in green colors) in comparison with mean diel variation of meteorological parameters (solar radiation, temperature displayed as red lines and orange boxes, respectively). This figure includes all measurement days with a PTR-MS (i.e. from 1 to 29 March 2015).**

**9/** For Figure 9, I have a very similar comment as for Figure 5. It is easier to understand the information if the different NPF event day classes are summarized as mean diel cycle Figure. Again, yellow on yellow.

We understand that it can be difficult to extract the information from Figure 9 (of the initial version of the manuscript), considering the number of parameters explored and the number of measurement days. As suggested by referee #2, mean diel variations of CS and $SO_2$ concentrations for the different NPF event day classes and for non-event days were added to Figure 11 (of the initial version of the manuscript – Figure 10 of the revised manuscript). Note that, diel variations of the selected parameters during NPF2 event days are now displayed in orange (instead of yellow) in Figure 10 (of the revised manuscript).

As a complement to Figure 10 (of the revised manuscript), Figure 9 (of the revised manuscript) presents mean diel variations of particle numbers ($N_{PSM}$, $N_{PSM-DMPS}$ and $N_{DMPS}$) and accumulated diel variations of $PM_1$ contributions.

Otherwise, Figure 9 (of the initial version of the manuscript) was kept in the revised version of the manuscript, but shifted to the Supplement as Figure SI-4. Considering the number of measurement days for DMPS and PSM (i.e. 20 days), the 4 NPF event day classes are at best represented by 4 event days. So the authors think that Figure SI-4 enables to study suspected parameter influences for each NPF event day individually, nuancing hence the statistical vision of the results given in Figures 9 and 10 (of the revised manuscript). For instance, according to Figure 10 (of the revised manuscript), high concentrations of monoterpenes seem to occur during the nights succeeding NPF2 events (i.e. 8-10 March and 23 March) but, according to Figure SI-4, high concentrations of monoterpenes were mainly observed during the night of the 10th of March. An additional importance of Figure SI-4 is the presentation of air mass origins.

Additionally, $H_2SO_4$ concentrations are presented in violet (instead of light orange) and NPF2 event days are depicted in orange (instead of yellow) in Figure SI-4.

Figures 9 and 10 of the revised manuscript are the followings:

[Figure]

**Figure 9: Diel variation of particle number $N_{PSM}$ and $N_{DMPS}$ and accumulated diel variations of $PM_1$ contributions for NPF event days (NPF1-NPF4) and non-event days. Diel variations are represented by daily mean values associated with standard deviation when several days were combined. Time is given in local time (UTC + 2 h).**

[Figure]

**Figure 10:** Diel variation of CS, SO₂, H₂SO₄, BVOCs (isoprene and monoterpenes) and meteorological parameters (global solar radiation, relative humidity and temperature) during NPF event days (NPF1-NPF4 displayed as red, orange, blue and green lines, respectively) and non-event days (grey lines). Diel variations are represented by daily mean values associated with standard deviation when several days were combined. Time is given in local time (UTC + 2 h).

Figure SI-4 in the Supplement is the following:

[Figure]

**Figure SI-4:** Time series of particle number $N_{PSM}$, $N_{DMPS}$ and CS in comparison with suspected parameters controlling NPF events (SO₂, H₂SO₄, isoprene and monoterpenes) and accumulated time series of PM₁ contribution. The color code highlights NPF event days and non-event days (grey periods). Red periods represent NPF1 event days with

**anthropogenic origin. Orange periods represent NPF2 event days both with mixed origins (anthropogenic and biogenic). Blue and green periods are respectively for NPF events of marine (NPF3) and biogenic origin (NPF4). Organic aerosol (OA) factors: HOA - hydrogen-like OA; SV-OOA – semi-volatile oxygen-like OA; LV-OOA – low-volatile oxygen-like OA. Time is given in local time (UTC + 2 h).**

**10/** I guess you chose the NPF2 in Figure 12, because of the high isoprene concentrations during that event class. I suggest again instead of showing time series of each day separately, to show mean diel cycle plots for the presented parameters comparing NPF2 event days and non-event days before or after NPF2 event days. Again, please avoid yellow on yellow.

The authors chose to further investigate 3 NPF2 event days (8-10 March – NPF event days of mixed origins) since $H_2SO_4$ and isoprene concentrations were particularly high during NPF2 event days (table 2 of the revised manuscript – authors' response 3) compared to others NPF event days. Similar diurnal variations were also observed between isoprene, temperature and $N_{DMPS-PSM}$ during NPF2 event days (Fig. 9 and 10 of the revised manuscript – authors' response 9) suggesting that isoprene and $H_2SO_4$ can both play a role during NPF2 event days.

Moreover, higher strength was noticed for NPF2 event day under mixed influence (anthropogenic and biogenic – 23 March) than the ones observed both during NPF1 and NPF4 event days under anthropogenic and biogenic origins respectively, for the same levels of precursors (anthropogenic and biogenic, respectively) suggesting that the combination of biogenic and anthropogenic species forms new compounds which may be involved in nucleation.

At similar levels of biogenic tracer, NPF2 event on 23 March was characterized by higher particulate formation and growth rates ($J_3$: 8.97 cm$^{-3}$s$^{-1}$ - $GR_{1.5-3}$: 3.18 nm.h$^{-1}$) compared to the mean rates characterizing the NPF4 event day of biogenic origin ($J_3$: 8.13 cm$^{-3}$s$^{-1}$ - $GR_{1.5-3}$: 1.93 nm.h$^{-1}$). This finding suggests polluted air mixed with high concentrations of biogenic tracers induced more intense particulate formation and faster growth. At higher $H_2SO_4$ and BVOC concentrations, NPF2 event days occurring on 8-10 March have shown higher particulate formation rates than the one of NPF event on 23 March ($J_3$: 12.23 cm$^{-3}$s$^{-1}$ in average ± 5.62 cm$^{-3}$s$^{-1}$ on 8-10 March and $J_3$: 8.97 cm$^{-3}$s$^{-1}$ on 20 March). This finding again, would confirm polluted air mixed with high concentrations of anthropogenic tracers can induce more intense particulate formation.

As a result, the Section 3.4.3 of the manuscript ("Focus on BVOC contributions to particle formation and growth") focuses on 8-10 March (mixed NPF event type) to better understand how the interaction of BVOC species with anthropogenic compounds can initiate nucleation and contributes to early growth of nucleated particles. This section is considered as a case study of 3 specific event days. These days had their specificities, that's why the authors do not prefer presenting results by mean diel cycles, that could biaise interpretations of variations of the selected parameters.

NPF2 event days were compared to non-event days in the previous section and in Figures 9-10 and Table 2 of the revised manuscript. To avoid any redundancy, the authors prefer not showing non-event days in Figure 11.

Additionally, $H_2SO_4$ concentrations are presented in violet (instead of light orange) in Figure 11 (of the revised manuscript) and orange color used for $NH_4$ and solar radiation has been darkened while yellow blocks have been lightened.

Figure 11 of the revised manuscript is the following:

[Figure]

[Figure]

**Figure 11:** Time series of $N_{PSM}$, $N_{DMPS}$, $N_{PSM-DMPS}$ and CS during NPF2 event days (i. e. 08-10 March) in comparison with meteorological parameters (global solar radiation, temperature, relative humidity and precipitation), $SO_2$, $H_2SO_4$, BVOCs (isoprene, MVK+MACR and monoterpenes) and $PM_1$ composition. Time is given in local time (UTC + 2 h). NPF events are represented in yellow and nighttime succeeding these NPF events are depicted in blue. These periods are discussed in Sect. 3.4.3.

---

## Editor Decision (ED1)

**acp-2018-297: "Driving parameters of biogenic volatile organic compounds and consequences on new particle formation observed at an Eastern Mediterranean background site"**

The manuscript by Debevec et al. describes VOC mixing ratios measured at Cyprus Atmospheric Observatory during one month, March 2015. The motivation for the paper is to find out the driving factors for new particle formation. The measurements cover both on-line and off-line measurements, altogether more than 60 compounds were detected. These kind of intensive campaigns are valuable, since there is still a lot of unknown reactive organic compounds in the atmosphere. Unfortunately, the measurements did not cover sesquiterpenes, since they are likely to be very important in new particle formation and there is very little ambient data of sesquiterpene mixing ratios. The paper is well written, it includes nice, informative figures and it suits well to be published in ACP after minor revisions described below.

**Authors' Responses to Referee #1**

We would like to thank the Referee #1 for her/his general feedback and each of her/his useful comments/questions for improving the quality of this manuscript. All comments addressed by both referees have been taken into account in the revised version of the manuscript. In this respect, several figures were notably modified included in the supplementary. Please note that figures numbers are now different in this new version.

In the present document, authors' answers to the specific comments addressed by Referee #1 are mentioned in **blue**, while changes made to the revised manuscript are shown in **green**.

The comments on the manuscript are listed as follows:

**1/** It is mentioned that ozone was removed in off-line sampling. $MnO_2$ removes also sesquiterpenes, which is unfortunate, since SQTs are likely to have an important contribution in SOA formation. How about ozone removal in BVOC on-line measurements? How was it removed? In the VOC intercomparison between the used methods on-line measurements showed lower values than off-line measurements. Could this be due to different ozone removal efficiency?

The intensive field campaign was initially carried out at the CAO to provide insights of the origins and fates of VOCs and aerosols in the Eastern Mediterranean, focusing on an extensive high time resolution in-situ measurements performed at a representative receptor site. Instruments deployed during this month make possible the measurement of many tracer compounds from various sources that have been observed at similar rural or remote sites (e.g. Lanz et al., 2008; Leuchner et al., 2015; Michoud et al., 2017; Sauvage et al., 2009; Vlasenko et al., 2009). Considering it was the first intensive field campaign realized at CAO, the authors did not expect to observe such elevated BVOC mixing ratios, especially in March, even if a biogenic potential was noticed considering the site is rather close of oak and pine forests. As a consequence, a specific GC sensitive enough to monitor sesquiterpenes was not deployed in this intensive field campaign. Additionally, sesquiterpenes can be measure with a PTR-MS as they can fragment partly at m/z 205 (Kim et al., 2009). Unfortunately, the

authors decided to limit the scan mode of the PTR-MS to m/z 137, in order to have a better time resolution and since no significant levels were noticed at higher m/z (check realized in late February 2015). In a future study, and following the suggestion of referee #1, it could be interesting to investigate sesquiterpene role in NPF formation at CAO.

We didn't use any ozone scrubber for on-line measurements (GCs and PTR-MS). However, as recommended by Detournay et al. (2011), different ozone scrubbers were used during the sampling of off-line measurements presented in section 2.2.1 in order to prevent any ozonolysis of the measured compounds. A KI ozone scrubber was installed upstream of the sampling onto DNPH cartridges, while a $MnO_2$ ozone scrubber was used for the multi-sorbent cartridges.

In addition to their on-line measurements, α-pinene and acetaldehyde were also measured by off-line techniques. α-Pinene was collected by multi-sorbent cartridges, analyzed after by GC-FID, while acetaldehyde was sampled with DNPH cartridges and analyzed by HPLC-UV. α-Pinene and acetaldehyde were hence chosen to see the potential influence of ozone on on-line measurement by the cross-checking of the results during the field campaign. On-line versus off-line measurements of α-pinene and acetaldehyde concentrations displayed a quite good correlation ($r^2$: 0.69 and 0.81 for α-pinene and acetaldehyde, respectively) and a slope close to one for both compounds (1.08 and 1.16 for α-pinene and acetaldehyde, respectively). Regarding these results, we think that potential interferences of ozone caused on VOCs measurements with GC systems were limited in this study.

As remarked by referee #1, on-line measurements of the sum of monoterpenes showed lower values than off-line measurements. The authors think that it could be partly due to β-pinene off-line measurements. The authors met some technical difficulties to correctly quantify β-pinene for the off-line method since the calibration results were not reproducible. It could notably affect the concentrations of the sum of monoterpenes for the off-line method, since β-pinene concentrations represented 32 % of the total monoterpene (i.e. the sum of 8 monoterpenes) concentrations.

Additionally, the authors consider that PTR-MS monoterpene measurements were more reliable than off-line monoterpene measurements considering their uncertainties (22 %) that's why on-line measurements were used in this study for the variability investigation of monoterpene concentrations.

**2/** It is not quite correct to say that monoterpenes were the most abundant group, when only monoterpenes, isoprene and few oxygenated compounds were measured.

Correction applied in the revised manuscript (Page 11, lines 1 – 2): *"Among BVOCs monitored during the intensive field campaign*, the most abundant were monoterpenes.*"

**3/** Was isoprene measured with PTR-MS only? As mentioned, also other compounds than isoprene can add to m/z 69, and it would be interesting to see a comparison of isoprene measurement in supplement with other comparisons. High nighttime isoprene concentrations could be due to other compounds fragments too. Inomata et al. conducted such a comparison (ACP; doi:10.5194/acp-10-7085-2010) and found that isoprene measurements with PTR-MS were overestimated in comparison with FID.

During the intensive field campaign, isoprene GC measurements were invalidated since we met technical problems (instability of the baseline around isoprene time response) which did not make possible a correct identification and quantification for isoprene as such levels. This problem was solved after the intensive field campaign, by the replacement of the $H_2$ generator by compressed $H_2$ in bottle which permitted to reduced baseline interferences. As a consequence, isoprene was monitored with a GC-FID from late April 2015 to October 2015 and hence comparison of isoprene PTR-MS and GC measurements cannot be done. The authors agree with referee #1 on the fact that nighttime isoprene concentrations could be due to other compounds fragments which was specified in Section 2.2.1 of the revised manuscript.

Correction applied in the revised manuscript (page 6, lines 2-4):
"Note that, nighttime isoprene concentrations discussed in Sect. 3.1 and 3.2 could be due to other compound fragments such as 2-methyl-3-butene-2-ol (MBO)."

**4/** It is not self-evident that MBO is temperature and light dependent in a same way as isoprene as mentioned. At least in boreal forest this could not be proved (Tarvainen et al., ACP doi.org/10.5194/acp-5-989-2005).

It was nuanced in the revised version of the manuscript as fallowed (Page 13, lines 4-7):
"In contrast to monoterpenes, The emissions of MBO could require light as isoprene (Harley et al., 1998) or could be mainly temperature dependent (Hellén et al., 2018; Tarvainen et al., 2005). Isoprene and/or MBO that were emitted during the late afternoon could be not fully oxidized photochemically, as OH concentrations begin to fall, and could remain in the nighttime atmosphere."

**5/** The chapter 3.2 has a misleading title. The manuscript deals with ambient mixing ratios, not with emissions. When discussing variability of the mixing ratios, the atmospheric mixing is not taken into consideration. There is a lot of discussion about the effect of humidity and rain in the mixing ratios, but these can be due to lower mixing layer height. There is currently also another paper under review in ACPD (Hellén et al., doi.org/10.5194/acp-2018-399) which claims that mixing layer height and temperature are the main factors determining ambient mixing ratios. Is there a way to evaluate mixing layer height at CAO if not measured? This would be extremely valuable and needs to be taken into account.

Correction applied in the revised manuscript (page 13, lines 8-13):
"3.2 Factors controlling BVOC concentrations

In this section, time variations of main monoterpenes and isoprene are examined along with meteorological parameters in order to determine the dominant factors controlling BVOC concentrations."

As unfortunately the Planet Boundary Layer (PBL) height was not measured at CAO, we used instead PBL assimilated data conducted for the Troodos station (32.88° E - 34.92° N, ~20 km westerly from the CAO station) which are described in the section S1 added in the Supplement. The comparison of BVOC time series with PBL height was added to the section 3.2 of the revised manuscript and consequently, Figure 5 was modified in the revised manuscript.

"2.2.4 Meteorological measurements and assimilated data

[revised manuscript text omitted]

The ERA-Interim dataset starts from 1979 and continues to provide information until present in near real-time. Gridded data products include a large variety of 3-hourly surface parameters, describing weather as well as ocean-wave and land-surface conditions, and 6-hourly upper-air parameters covering the troposphere and stratosphere. Vertical integrals of atmospheric fluxes, monthly averages for many of the parameters, and other derived fields have also been produced. Berrishford et al. (2011) provide a detailed description of the ERA-Interim product archive. ERA-Interim products are normally updated once per month, with a delay of two months to allow for quality assurance and for correcting technical problems. The ERA-Interim atmospheric model has a spatial resolution of 0.75°x0.75° and expands vertically with 60 atmospheric layers. The reanalysis product is produced with a sequential data assimilation scheme, using 12-hourly analysis cycles , a time-window when available observations are assimilated into the information from the forecast model as described in Dee et al. (2011).

The ERA-Interim model includes a PBL height parameter calculated from the Bulk Richardson number (Troen and Mahrt, 1986), which is based on ratios of both dynamic and thermodynamic vertical gradients and hence characterizes the degree of turbulence. Given the fact that the boundary layer is often associated with stronger mixing (as compared to the free troposphere) due to increased levels of turbulence, it would be natural to investigate properties associated with turbulence. Essentially, the PBL height is defined as the level where the bulk Richardson number reaches a critical value of 0.25, based on the difference between quantities at this level and the lowest model level as an estimator for the vertical stability. Bulk Richardson number is available a 6-h and 12-h forecasts.

However, as reported in von Engeln and Teixeira (2013) this method of estimating the stability from dry thermodynamic variables (not moist), tends to provide estimates of PBL height that is often closer to the cloud-base height in marine cloudy boundary layers, rather than the PBL height itself (Janssen and Bidlot, 2003). Seidel et al. (2012) reports that for their scope of assessing the climatology of the planetary boundary layer over the continental United States and Europe with the use of ERA-Interim datasets, they did not employ the estimates of the BLH from ERA-Interim itself, because they are computed using an algorithm not applicable to radiosonde data (due to the fact that turbulence parameters are required for this application). With a preliminary analysis, they report that the ERA-interim PBL height product (i.e., with the ECMWF algorithm) shows higher

heights, especially over high elevation regions, than the algorithm used in their study on the radiosonde data. Differences were below 100 m at night and of several 100 m during daytime."

Revised Figure 5 is the following:

[Figure]

[Figure]

**Figure 5: Time series of isoprene and a selection of monoterpenes (α-pinene and β-pinene) in comparison with time series of meteorological parameters (boundary layer height, wind speed, solar radiation, temperature, precipitation and relative humidity). Blue rectangles correspond to nighttime periods. BVOC episodes 1 to 5 referred to specific BVOC variations discussed in Sect. 3.2. Note that, PBL assimilated data were generated by the ECMWF Era-Interim global atmospheric reanalysis at the location corresponding to the Troodos station (32.88° E – 34.92° N, ~20 km westerly from the CAO station).**

**6/** Acetaldehyde is a known product of myriad of atmospheric reactions, for example OH radical reactions. OH radicals are produced only in sunlight and therefore acetaldehyde mixing ratios would also peak during daytime. The following sentence is from Millet et al. (ACP, 2010) abstract: "Hydrocarbon oxidation provides the largest acetaldehyde source in the model (128 Tg a-1, a factor of 4 greater than the previous estimate), with alkanes, alkenes, and ethanol the main precursors. There is also a minor source from isoprene oxidation". Why are the authors convinced that VOC oxidation is not the cause for high midday acetaldehyde mixing ratios, but light dependent emissions?

From the 6 PMF factors reported in Debevec et al. (2017), the measured OVOCs were distributed among their different sources (Fig. 6). More than 80 % of the respective total mass of methanol, acetaldehyde and MVK+MACR was explained by biogenic sources, namely by factor 2 driven by isoprene emissions. The morning increase pattern of acetaldehyde concentration is similar to isoprene pattern rather than isoprene oxidation product one that could suggest that acetaldehyde was mostly released in the atmosphere by local vegetation rather than produced by VOC oxidation processes in March 2015.

Correction applied in the revised manuscript (Page 16, lines 15-24):
"Hydrocarbon oxidation (mostly alkanes and alkenes but also isoprene and ethanol) provides the largest acetaldehyde source in the budget estimates of Millet et al. (2010). Nonetheless, for all reaction pathways of isoprene with atmospheric oxidants, acetaldehyde is produced as a second- or higher-generation oxidation product of isoprene (Millet et al., 2010). In addition to photochemical production, acetaldehyde is emitted by terrestrial plants, as a result of fermentation reactions leading to ethanol production in leaves and roots (Jardine et al., 2008; Rottenberger et al., 2008; Winters et al., 2009).  Acetaldehyde concentrations started to increase in the morning and peaked at

midday followed by a gradual decrease throughout the rest of the day. The morning increase pattern of acetaldehyde concentrations is similar to the isoprene pattern rather than isoprene oxidation product one and isoprene and acetaldehyde also correlated well ($r^2$ = 0.49). These findings suggesting that acetaldehyde was mostly released in the atmosphere by local vegetation rather than produced by VOC oxidation processes in March 2015.

**7/** Table showing the mean mixing ratios of individual compounds would be helpful

The table showing statistics of mixing ratios of individual compounds was added to the Supplement of the revised manuscript as Table SI-1.

Table SI-1 in the Supplement is the following:

**Table S1: Statistics (µg.m$^{-3}$), detection limits (DL - µg.m$^{-}$3) and relative uncertainties u(X)/X (Unc. - %) of selected VOC concentrations measured at the site.**

|  | **Species** | **Min** | **25 %** | **50 %** | **Mean** | **75 %** | **Max** | **σ** | **DL** | **Unc.** |
|---|---|---|---|---|---|---|---|---|---|---|
| **DIENE** | **Isoprene** | 4 | 26 | 38 | 46 | 53 | 219 | 28 | 21 | 11 |
|  |  |  |  |  |  |  |  |  |  |  |
| **TERPENES** | **α-Pinene** | 8 | 8 | 18 | 58 | 58 | 1874 | 131 | 16 | 10 |
|  | **β-Pinene** | 6 | 6 | 18 | 61 | 57 | 1962 | 142 | 12 | 12 |
|  | **Camphene** | <1 | 5 | 11 | 25 | 29 | 275 | 37 | 1 | ND |
|  | **Myrcene** | <1 | 2 | 4 | 6 | 8 | 43 | 7 | 2 | ND |
|  | **Δ$^3$-Carene** | <1 | 4 | 8 | 11 | 15 | 91 | 11 | 1 | ND |
|  | **α-Terpinene** | <1 | 1 | 2 | 3 | 5 | 32 | 4 | 1 | ND |
|  | **γ-Terpinene** | <1 | <1 | <1 | <1 | 1 | 12 | 2 | 1 | ND |
|  | **Limonene** | <1 | 8 | 17 | 27 | 32 | 347 | 37 | 1 | ND |
|  |  |  |  |  |  |  |  |  |  |  |
| **ALCOHOL** | **Methanol** | 654 | 1658 | 2426 | 2765 | 3452 | 9074 | 1452 | 180 | 21 |
|  |  |  |  |  |  |  |  |  |  |  |
| **CARBONYL COMPOUNDS** | **Formaldehyde** | 399 | 678 | 909 | 986 | 1170 | 2416 | 409 | 25 | ND |
|  | **Acetaldehyde** | 102 | 277 | 390 | 431 | 531 | 1533 | 209 | 44 | 10 |
|  | **Acetone** | 423 | 861 | 1048 | 1083 | 1214 | 2662 | 335 | 17 | 9 |
|  | **MVK+MACR** | 3 | 19 | 26 | 30 | 35 | 139 | 18 | 3 | 12 |
|  | **MEK** | 59 | 154 | 196 | 210 | 242 | 653 | 84 | 13 | 9 |

ND: not determined

**acp-2018-297: "Driving parameters of biogenic volatile organic compounds and consequences on new particle formation observed at an Eastern Mediterranean background site"**

The presented manuscript describes the on-line and off-line measurements of various organic compounds at a remote Mediterranean measurement site. The measurements include 20 days of data. The authors present characteristics of 4 different NPF classes, which they categorized based on air mass origin. The manuscript presents very interesting new results. I suggest minor revisions, described in the following.

**Authors' Responses to Referee #2**

We would like to thank the Referee #2 for her/his general feedback and each of her/his useful comments/questions for improving the quality of this manuscript. All comments addressed by both referees have been taken into account in the revised version of the manuscript. In this respect, several figures were notably modified and included in the supplementary. Please note that figures numbers are now different in this new version.

In the present document, authors' answers to the specific comments addressed by Referee #2 are mentioned in **blue**, while changes made into the revised manuscript are shown in **green**.

The comments on the manuscript are listed as follows:

**1/** About the writing style of the manuscript, there are quite a lot of grammatical mistakes in the manuscript and it is very difficult to read. I suggest asking a native English speaker to correct the language before re-submitting.

The revised manuscript was corrected by a native English speaker. The referee #2 is invited to look at the peer review version of the revised manuscript in order to see all the modifications made consequently to his/her comment.

**2/** Why are you not showing any data from the NAIS measurements? It would be very interesting to see mean diel cycles for different size classes below 20 nm from the NAIS measurements for different NPF event day classes and non-event days. A comparison to PSM size classes and DMPS would be helpful in the same figures.

In this study, daily size distribution spectra measured with NAIS were mainly used to strengthen the identification and the classification of NPF events. In fact, the authors followed the classification scheme of Yli-Juuti et al. (2009), combining visual observation of NPF events from (N)AIS and DMPS measurements. The evolution of particle size distributions also gives us a way to know their growth and nucleation rates.

Additionally, during the intensive field campaign, the PSM was not operated in the scan mode (for the measurement of all particles having a diameter between 1 and 2.5 nm) but in the

total mode (for the measurement of all particles larger than 1 nm), which did not allow any growth and nucleation rate calculation in the size range 1-3 nm.

As mentioned by referee #2, there are already numerous Figures containing many information that's why the authors prefer not showing NAIS mean diel cycles in this study. Moreover, the aim of this study is not to provide an extensive investigation of NPF events but rather to focus on the role of BVOCs in the early stages of formation and the growth of atmospheric aerosol particles.

In the preliminary study, each DMPS size class was investigated individually. During NPF events, number concentrations of larger size particle classes can increase with a delay compared to number concentrations of first particle size class (20-27 nm from 8 to 11 March and 10-13 nm from 12 to 27 March). An example is provided in Figure X1, in agreement with the banana-shape depicted in Figure 8 of the manuscript.

Furthermore, to summarize the results and according to the aim of this study, the authors only made the distinction between $N_{PSM-DMPS}$, corresponding to number concentrations of sub-20 nm particles (from 8 to 11 March) or sub-10 nm particles (from 12 to 27 March) and $N_{DMPS}$, corresponding to number concentrations of either 20-200 nm particles (from 8 to 11 March) or 10-250 nm particles (from 12 to 27 March) in the manuscript. These two parameters can provide information on the early stages of formation (regarding $N_{PSM-DMPS}$) and the growth ($N_{DMPS}$) of atmospheric aerosol particles. As a result, $N_{PSM}$ was decomposed into $N_{DMPS}$ and $N_{PSM-DMPS}$ in Figure 9 of the revised manuscript (see authors' response 9). The authors hope that Figure 9 of the revised manuscript will meet referee #2 expectations about comparisons of PSM and DMPS measurements in the same Figures.

[Figure]

**Figure 8: Example of size distribution spectra, measured with DMPS and NAIS, showing an NPF event of type Ia occurring on 14 March 2015 at the CAO station**

[Figure]

**Figure X1: Time series of a selection of number concentrations, measured with the DMPS, showing an NPF event of type Ia occurring on 14 March 2015 at the CAO station**

**3/** A table, summarizing the findings regarding NPF event days and non-event days is needed. That table could contain the information that is shown in Figures 10 and 11, for the different NPF classes found in your analysis.

As proposed by referee #2, Figure 10 (of the initial version of the manuscript) was removed and converted into a table (as Table 2 in the revised manuscript) showing mean and standard deviation values for atmospheric parameters (supporting the classification of event days) along with property indicators for NPF events and factors with suspected influence on nucleation events.

Otherwise, Figure 11 was kept, and hence not integrated to Table 2, since the importance of diurnal variations as point out by referee #2 (please see authors' response 9 in complement). Nevertheless, mean and standard deviation values for some meteorological parameters (temperature, relative humidity and solar radiation) were added to Table 2 of the revised manuscript.

Table 2 of the revised manuscript is the following:

**Table 2: Average and standard deviation of CS, particle formation and growth rates ($J_{1.5}$ and $GR_{1.5-3}$, respectively), meteorological parameters (temperature, relative humidity and solar radiation) and atmospheric parameters daily concentrations measured at the CAO station in case of event (NPF1-NPF4) or non-event days.**

| Parameter | NPF1 event days | NPF2 event days | NPF3 event days | NPF4 event days | Non-event days |
|---|---|---|---|---|---|
| CS ($s^{-1}$) | $0.12 \pm 0.02$ | $0.09 \pm 0.02$ | $0.08 \pm 0.01$ | 0.12 | $0.07 \pm 0.04$ |
| $J_3$ ($cm^{-3}s^{-1}$) | 5.0 | $11.4 \pm 4.9$ | $6.4 \pm 1.4$ | 8.1 | - |
| $GR_{1.5-3}$ ($nm.h^{-1}$) | 5.0 | $3.7 \pm 1.6$ | $1.9 \pm 0.6$ | 2.8 | - |
| $PM_1$ ($\mu g.m^{-3}$) | $9.7 \pm 1.4$ | $12.9 \pm 2.8$ | $5.9 \pm 0.7$ | 9.8 | $6.4 \pm 3.6$ |
| $SO_4$ ($\mu g.m^{-3}$) | $2.9 \pm 0.7$ | $3.3 \pm 1.0$ | $1.9 \pm 0.3$ | 3.1 | $1.9 \pm 1.3$ |
| $NH_4$ ($\mu g.m^{-3}$) | $1.9 \pm 0.4$ | $2.1 \pm 0.6$ | $1.2 \pm 0.2$ | 1.8 | $1.2 \pm 0.8$ |
| $NO_3$ ($\mu g.m^{-3}$) | $0.5 \pm 0.2$ | $0.7 \pm 0.2$ | $0.3 \pm 0.1$ | 0.3 | $0.3 \pm 0.1$ |
| OM ($\mu g.m^{-3}$) | $4.3 \pm 0.4$ | $6.8 \pm 1.5$ | $2.6 \pm 0.3$ | 4.5 | $2.9 \pm 1.5$ |
| HOA ($\mu g.m^{-3}$) | $0.4 \pm 0.1$ | $0.7 \pm 0.2$ | $0.3 \pm 0.1$ | 0.4 | $0.3 \pm 0.1$ |
| SV-OOA ($\mu g.m^{-3}$) | $1.3 \pm 0.2$ | $2.5 \pm 0.9$ | $0.8 \pm 0.2$ | 1.1 | $0.9 \pm 0.4$ |
| LV-OOA ($\mu g.m^{-3}$) | $1.7 \pm 0.2$ | $1.8 \pm 0.5$ | $1.3 \pm 0.1$ | 2.2 | $1.3 \pm 0.7$ |
| BC ($\mu g.m^{-3}$) | $0.5 \pm 0.1$ | $1.0 \pm 0.3$ | $0.3 \pm 0.1$ | 0.4 | $0.3 \pm 0.1$ |
| CO (ppb) | $158.2 \pm 5.5$ | $162.5 \pm 9.2$ | $160.1 \pm 19.5$ | 155.1 | $151.6 \pm 13.2$ |
| $NO_2$ (ppb) | $1.1 \pm 0.2$ | $1.4 \pm 0.5$ | $0.8 \pm 0.1$ | 0.7 | $0.6 \pm 0.2$ |
| $SO_2$ (ppb) | $0.7 \pm 0.3$ | $0.7 \pm 0.3$ | $0.3 \pm 0.1$ | 0.2 | $0.2 \pm 0.1$ |
| $H_2SO_4$ ($molec.cm^{-3}$) | $6.3\ 10^7 \pm 5.2\ 10^7$ | $1.4\ 10^8 \pm 8.4\ 10^7$ | $4.3\ 10^7 \pm 1.8\ 10^7$ | $1.8\ 10^7$ | $2.3\ 10^7 \pm 1.7\ 10^7$ |
| Isoprene (ppt) | $34 \pm 7$ | $79 \pm 29$ | $33 \pm 7$ | 57 | $47 \pm 16$ |
| MVK+MACR (ppt) | $27 \pm 4$ | $61 \pm 23$ | $25 \pm 1$ | 26 | $30 \pm 8$ |
| Monoterpenes (ppt) | $115 \pm 19$ | $361 \pm 209$ | $148 \pm 80$ | 130 | $306 \pm 204$ |
| $O_3$ (ppb) | $50.4 \pm 3.7$ | $48.2 \pm 2.8$ | $46.4 \pm 2.6$ | 48.2 | $46.5 \pm 4.3$ |
| Temperature (°C) | $14.2 \pm 2.4$ | $15.4 \pm 3.7$ | $11.8 \pm 2.4$ | 10.7 | $11.2 \pm 1.7$ |
| Relative Humidity (%) | $54.0 \pm 12.3$ | $63.5 \pm 18.1$ | $61.3 \pm 9.6$ | 63.8 | $79.6 \pm 12.5$ |
| Solar radiation ($W.m^{-2}$) | $258 \pm 213$ | $255 \pm 192$ | $305 \pm 228$ | 283 | $203 \pm 199$ |

**4/** The presented Figures are extensive and contain a lot of information. Please do not use yellow in your Figures, it is very hard to read the content of the Figures if there are yellow lines.

An effort was realized to limit the use of yellow/light orange colors in the Figures of the manuscript. As a consequence, the color used to represent NPF event days categorized by a mixed (anthropogenic/biogenic) influence is now a dark orange (instead of yellow) in order to stay consistent with colors used for NPF event days of individual origin (i.e. red for NPF1 event days of anthropogenic origin and green for NPF3 event days of biogenic origin). The orange color used to represent solar radiation and $NH_4$ data has been darkened and $H_2SO_4$ concentrations are now represented in violet (instead of light orange).

The modifications applied to Figures 5, 9, 10 and 11 (of the initial version of manuscript) are explicit in the following answers.

**5/** It is sometimes difficult to extract all the information in the Figures. I will make some detailed suggestions in the following.

The authors thank referee #2 for these detailed suggestions which the authors will take into account in the following.

**6/** In Figure 4, it is not clear to me, what exactly is presented here? Do those Figures include all measurement days, NPF event days only or non-event days only? Please do not use yellow.

In Figure 4 is presented diurnal variations of isoprene and monoterpenes concentrations. These diurnal variations are also compared with mean diel variations of meteorological parameters (temperature and solar radiation) which are known to influence BVOC emissions, and so indirectly BVOC concentration variations.

This figure includes all BVOC measurement days with a PTR-MS, i.e. from 1 March to 29 March 2015, which has been specified in the caption of Figure 4. This period includes NPF event days and non-event days as the variation of BVOC concentrations was independent of this element.

As suggest by referee #2, the orange color used to represent solar radiation data has been darkened.

Revised Figure 4 is the following:

[Figure]

**Figure 4: Diel variation of isoprene and monoterpenes, represented by hourly box plots (in green colors) in comparison with mean diel variation of meteorological parameters (solar radiation, temperature displayed as red lines and orange boxes, respectively). This figure includes all BVOC measurement days with a PTR-MS (i.e. from 1 to 29 March 2015). White marker represents the mean value, blue solid line represents the median value and the green**

**box shows the InterQuartile Range (IQR). The bottom and the top of box depict the first and the third quartiles (i.e. Q1 and Q3). The ends of the whiskers correspond to the first and the ninth deciles (i.e. D1 and D9). Time is given in local time (UTC + 2 h).**

**7/** Figure 5 is very difficult to read, there is yellow on yellow and an extensive amount of information. I suggest making mean diel cycle Figures, summarizing the different NPF event day classes you observed, showing the same parameters as in each panel of the current Figure.

In Figure 5 is presented times variations of main monoterpenes and isoprene examined along with meteorological parameters in order to determine the dominant drivers for variations of BVOC concentrations.

      Given the extensive amount of information for the Figure 5, the investigation of meteorological parameter effects on BVOC concentrations was mainly based on the study of 5 specific periods among the 29 days of BVOC measurements, called "events" in the initial version of manuscript. Otherwise, the appellation of "event" does not refer to NPF event day. Thanks to referee #2 comment, the authors realized that the use of the term "event" in this section can lead to confusion. As a result, the 5 specific periods are now called "episodes" in the revised manuscript.

      The suggestion of referee #2 in making mean diel cycle Figures, summarizing the different NPF event day classes observed is relevant. The authors hope that Figure 10 of the revised manuscript (see authors' response 9) showing mean diel variations for some meteorological parameters (solar radiation, relative humidity and temperature) and BVOCs (isoprene and monoterpenes) among others meets referee #2's expectations on this point.

      As suggested by referee #2, the orange color used to represent solar radiation data has been darkened.

Figure 5 of the revised manuscript is the following:

[Figure]

[Figure]

**Figure 5:** Time series of isoprene and a selection of monoterpenes (α-pinene and β-pinene) in comparison with time series of meteorological parameters (boundary layer height, wind speed, solar radiation, temperature, precipitation and relative humidity). Blue rectangles correspond to nighttime periods. BVOC episodes 1 to 5 referred to specific BVOC variations discussed in Sect. 3.2. Note that, PBL assimilated data were generated by the ECMWF Era-Interim global atmospheric reanalysis at the location corresponding to the Troodos station (32.88° E – 34.92° N, ~20 km westerly from the CAO station).

**8/** Figure 7 again, please avoid yellow. I do not really understand the difference between the first and the second panel, other than the second panel shows the same information as Panel 1, with added Methanol diel cycle. Maybe those two can be summarized in one panel? If there is a good reason to keep the first two panels separated, please explain it somewhere. I am not sure, which days are summarized here? Does that Figures include all measurement days? NPF event days, non-event days?

The first panel of Figure 7 highlights the delay of about 1 hour in the peak values between isoprene and its first oxidation products (MVK+MACR). On the second panel, BVOC concentrations are scaled differently than on the first one, which may make less obvious the occurrence of this delay. Considering the recommendation of referee #2, the panel 1 was moved to the Supplement (as Figure SI-3), in order to make Figure 7 of the revised manuscript less extensive.

Figure 7 includes all measurement days with a PTR-MS, i.e. from 1 March to 29 March 2015, which has been explicit in its caption. This period includes NPF event days and non-event days since the variation of acetaldehyde and methanol concentrations was studied independently from this element.

Figure 7 of the revised manuscript is the following:

[Figure]

**Figure 7: Diel variation of methanol and acetaldehyde, represented by hourly box plots (in blue colors) in comparison with mean diel variation of meteorological parameters (solar radiation, temperature displayed as red lines and orange boxes, respectively) and isoprene and its oxidation products (in green colors). This figure includes all measurement days with a PTR-MS (i.e. from 1 to 29 March 2015). White marker represents the mean value, blue solid line represents the median value and the green box shows the interquartile range. The bottom and the top of box depict the first and the third quartiles (i.e. Q1 and Q3). The ends of the whiskers correspond to the first and the ninth deciles (i.e. D1 and D9). Time is given in local time (UTC + 2 h).**

Figure SI-3 of the Supplement is the following:

[Figure]

**Figure SI-3: Mean diel variation of isoprene and its oxidation products (in green colors) in comparison with mean diel variation of meteorological parameters (solar radiation, temperature displayed as red lines and orange boxes, respectively). This figure includes all measurement days with a PTR-MS (i.e. from 1 to 29 March 2015).**

**9/** For Figure 9, I have a very similar comment as for Figure 5. It is easier to understand the information if the different NPF event day classes are summarized as mean diel cycle Figure. Again, yellow on yellow.

We understand that it can be difficult to extract the information from Figure 9 (of the initial version of the manuscript), considering the number of parameters explored and the number of measurement days. As suggested by referee #2, mean diel variations of CS and $SO_2$ concentrations for the different NPF event day classes and for non-event days were added to Figure 11 (of the initial version of the manuscript – Figure 10 of the revised manuscript). Note that, diel variations of the selected parameters during NPF2 event days are now displayed in orange (instead of yellow) in Figure 10 (of the revised manuscript).

As a complement to Figure 10 (of the revised manuscript), Figure 9 (of the revised manuscript) presents mean diel variations of particle numbers ($N_{PSM}$, $N_{PSM-DMPS}$ and $N_{DMPS}$) and accumulated diel variations of $PM_1$ contributions.

Otherwise, Figure 9 (of the initial version of the manuscript) was kept in the revised version of the manuscript, but shifted to the Supplement as Figure SI-4. Considering the number of measurement days for DMPS and PSM (i.e. 20 days), the 4 NPF event day classes are at best represented by 4 event days. So the authors think that Figure SI-4 enables to study suspected parameter influences for each NPF event day individually, nuancing hence the statistical vision of the results given in Figures 9 and 10 (of the revised manuscript). For instance, according to Figure 10 (of the revised manuscript), high concentrations of monoterpenes seem to occur during the nights succeeding NPF2 events (i.e. 8-10 March and 23 March) but, according to Figure SI-4, high concentrations of monoterpenes were mainly observed during the night of the 10th of March. An additional importance of Figure SI-4 is the presentation of air mass origins.

Additionally, $H_2SO_4$ concentrations are presented in violet (instead of light orange) and NPF2 event days are depicted in orange (instead of yellow) in Figure SI-4.

Figures 9 and 10 of the revised manuscript are the followings:

[Figure]

**Figure 9: Diel variation of particle number $N_{PSM}$ and $N_{DMPS}$ and accumulated diel variations of $PM_1$ contributions for NPF event days (NPF1-NPF4) and non-event days. Diel variations are represented by daily mean values associated with standard deviation when several days were combined. Time is given in local time (UTC + 2 h).**

[Figure]

[Figure]

**Figure 10: Diel variation of CS, SO$_2$, H$_2$SO$_4$, BVOCs (isoprene and monoterpenes) and meteorological parameters (global solar radiation, relative humidity and temperature) during NPF event days (NPF1-NPF4 displayed as red, orange, blue and green lines, respectively) and non-event days (grey lines). Diel variations are represented by daily mean values associated with standard deviation when several days were combined. Time is given in local time (UTC + 2 h).**

Figure SI-4 in the Supplement is the following:

[Figure]

**Figure SI-4:** Time series of particle number $N_{PSM}$, $N_{DMPS}$ and CS in comparison with suspected parameters controlling NPF events ($SO_2$, $H_2SO_4$, isoprene and monoterpenes) and accumulated time series of $PM_1$ contribution. The color code highlights NPF event days and non-event days (grey periods). Red periods represent NPF1 event days with anthropogenic origin. **Orange** periods represent NPF2 event days both with mixed origins (anthropogenic and biogenic). Blue and green periods are respectively for NPF events of marine (NPF3) and biogenic origin (NPF4). Organic aerosol (OA) factors: HOA - hydrogen-like OA; SV-OOA – semi-volatile oxygen-like OA; LV-OOA – low-volatile oxygen-like OA. Time is given in local time (UTC + 2 h).

**10/** I guess you chose the NPF2 in Figure 12, because of the high isoprene concentrations during that event class. I suggest again instead of showing time series of each day separately, to show mean diel cycle plots for the presented parameters comparing NPF2 event days and non-event days before or after NPF2 event days. Again, please avoid yellow on yellow.

The authors chose to further investigate 3 NPF2 event days (8-10 March – NPF event days of mixed origins) since $H_2SO_4$ and isoprene concentrations were particularly high during NPF2 event days (table 2 of the revised manuscript – authors' response 3) compared to others NPF event days. Similar diurnal variations were also observed between isoprene, temperature and $N_{DMPS-PSM}$ during NPF2 event days (Fig. 9 and 10 of the revised manuscript – authors' response 9) suggesting that isoprene and $H_2SO_4$ can both play a role during NPF2 event days.

Moreover, higher strength was noticed for NPF2 event day under mixed influence (anthropogenic and biogenic – 23 March) than the ones observed both during NPF1 and NPF4 event days under anthropogenic and biogenic origins respectively, for the same levels of precursors (anthropogenic and biogenic, respectively) suggesting that the combination of biogenic and anthropogenic species forms new compounds which may be involved in nucleation.

At similar levels of biogenic tracer, NPF2 event on 23 March was characterized by higher particulate formation and growth rates ($J_3$: 8.97 cm$^{-3}$s$^{-1}$ - $GR_{1.5-3}$: 3.18 nm.h$^{-1}$) compared to the mean rates characterizing the NPF4 event day of biogenic origin ($J_3$: 8.13 cm$^{-3}$s$^{-1}$ - $GR_{1.5-3}$: 1.93 nm.h$^{-1}$). This finding suggests polluted air mixed with high concentrations of biogenic tracers induced more intense particulate formation and faster growth. At higher $H_2SO_4$ and BVOC concentrations, NPF2 event days occurring on 8-10 March have shown higher particulate formation rates than the one of NPF event on 23 March ($J_3$: 12.23 cm$^{-3}$s$^{-1}$ in average ± 5.62 cm$^{-3}$s$^{-1}$ on 8-10 March and $J_3$: 8.97 cm$^{-3}$s$^{-1}$ on 20 March). This finding again, would confirm polluted air mixed with high concentrations of anthropogenic tracers can induce more intense particulate formation.

As a result, the Section 3.4.3 of the manuscript ("Focus on BVOC contributions to particle formation and growth") focuses on 8-10 March (mixed NPF event type) to better understand how the interaction of BVOC species with anthropogenic compounds can initiate nucleation and contributes to early growth of nucleated particles. This section is considered as a case study of 3 specific event days. These days had their specificities, that's why the authors do not prefer presenting results by mean diel cycles, that could biaise interpretations of variations of the selected parameters.

NPF2 event days were compared to non-event days in the previous section and in Figures 9-10 and Table 2 of the revised manuscript. To avoid any redundancy, the authors prefer not showing non-event days in Figure 11.

Additionally, $H_2SO_4$ concentrations are presented in violet (instead of light orange) in Figure 11 (of the revised manuscript) and orange color used for $NH_4$ and solar radiation has been darkened while yellow blocks have been lightened.

Figure 11 of the revised manuscript is the following:

[revised manuscript text omitted]

**Section S1 Planet boundary layer (PBL) assimilated data**

In order to investigate PBL height effect on BVOC concentrations, this parameter was evaluated using PBL assimilated data generated by the European Centre for Medium-Range Weather Forecast (ECMWF) Interim Re-Analysis (ERA-Interim) global atmospheric reanalysis at the location corresponding to the Troodos station (32.88° E - 34.92° N, ~20 km westerly from the CAO station).

The ERA-Interim dataset starts from 1979 and continues to provide information until present in near real-time. Gridded data products include a large variety of 3-hourly surface parameters, describing weather as well as ocean-wave and land-surface conditions, and 6-hourly upper-air parameters covering the troposphere and stratosphere. Vertical integrals of atmospheric fluxes, monthly averages for many of the parameters, and other derived fields have also been produced. Berrishford et al. (2011) provide a detailed description of the ERA-Interim product archive. ERA-Interim products are normally updated once per month, with a delay of two months to allow for quality assurance and for correcting technical problems. The ERA-Interim atmospheric model has a spatial resolution of 0.75°x0.75° and expands vertically with 60 atmospheric layers. The reanalysis product is produced with a sequential data assimilation scheme, using 12-hourly analysis cycles , a time-window when available observations are assimilated into the information from the forecast model as described in Dee et al. (2011).

The ERA-Interim model includes a PBL height parameter calculated from the Bulk Richardson number (Troen and Mahrt, 1986), which is based on ratios of both dynamic and thermodynamic vertical gradients and hence characterizes the degree of turbulence. Given the fact that the boundary layer is often associated with stronger mixing (as compared to the free troposphere) due to increased levels of turbulence, it would be natural to investigate properties associated with turbulence. Essentially, the PBL height is defined as the level where the bulk Richardson number reaches a critical value of 0.25, based on the difference between quantities at this level and the lowest model level as an estimator for the vertical stability. Bulk Richardson number is available a 6-h and 12-h forecasts.

However, as reported in von Engeln and Teixeira (2013) this method of estimating the stability from dry thermodynamic variables (not moist), tends to provide estimates of PBL height that is often closer to the cloud-base height in marine cloudy boundary layers, rather than the PBL height itself (Janssen and Bidlot, 2003). Seidel et al. (2012) reports that for their scope of assessing the climatology of the planetary boundary layer over the continental United States and Europe with the use of ERA-Interim datasets, they did not employ the estimates of the BLH from ERA-Interim itself, because they are computed using an algorithm not applicable to radiosonde data (due to the fact that turbulence parameters are required for this application). With a preliminary analysis, they report that the ERA-interim PBL height product (i.e., with the ECMWF algorithm) shows higher heights, especially over high elevation regions, than the algorithm used in their study on the radiosonde data. Differences were below 100 m at night and of several 100 m during daytime.

**Table S1: Statistics (µg.m$^{-3}$), detection limits (DL - µg.m$^{-}$3) and relative uncertainties u(X)/X (Unc. - %) of selected VOC concentrations measured at the site.**

|  | Species | Min | 25 % | 50 % | Mean | 75 % | Max | σ | DL | Unc. |
|---|---|---|---|---|---|---|---|---|---|---|
| **DIENE** | **Isoprene** | 4 | 26 | 38 | 46 | 53 | 219 | 28 | 21 | 11 |
|  |  |  |  |  |  |  |  |  |  |  |
| **TERPENES** | **α-Pinene** | 8 | 8 | 18 | 58 | 58 | 1874 | 131 | 16 | 10 |
|  | **β-Pinene** | 6 | 6 | 18 | 61 | 57 | 1962 | 142 | 12 | 12 |
|  | **Camphene** | <1 | 5 | 11 | 25 | 29 | 275 | 37 | 1 | ND |
|  | **Myrcene** | <1 | 2 | 4 | 6 | 8 | 43 | 7 | 2 | ND |
|  | **Δ$^3$-Carene** | <1 | 4 | 8 | 11 | 15 | 91 | 11 | 1 | ND |
|  | **α-Terpinene** | <1 | 1 | 2 | 3 | 5 | 32 | 4 | 1 | ND |
|  | **γ-Terpinene** | <1 | <1 | <1 | <1 | 1 | 12 | 2 | 1 | ND |
|  | **Limonene** | <1 | 8 | 17 | 27 | 32 | 347 | 37 | 1 | ND |
|  |  |  |  |  |  |  |  |  |  |  |
| **ALCOHOL** | **Methanol** | 654 | 1658 | 2426 | 2765 | 3452 | 9074 | 1452 | 180 | 21 |
|  |  |  |  |  |  |  |  |  |  |  |
| **CARBONYL COMPOUNDS** | **Formaldehyde** | 399 | 678 | 909 | 986 | 1170 | 2416 | 409 | 25 | ND |
|  | **Acetaldehyde** | 102 | 277 | 390 | 431 | 531 | 1533 | 209 | 44 | 10 |
|  | **Acetone** | 423 | 861 | 1048 | 1083 | 1214 | 2662 | 335 | 17 | 9 |
|  | **MVK+MACR** | 3 | 19 | 26 | 30 | 35 | 139 | 18 | 3 | 12 |
|  | **MEK** | 59 | 154 | 196 | 210 | 242 | 653 | 84 | 13 | 9 |

ND: not determined

**Figure S1: Agreement between on-line and off-line measurements of α-pinene, β-pinene and the sum of monoterpenes**

[Figure]

**Figure S2: Agreement between on-line and off-line measurements of acetaldehyde, acetone and MEK**

[Figure]

**Figure S3: Mean diel variation of isoprene and its oxidation products (in green colors) in comparison with mean diel variation of meteorological parameters (solar radiation, temperature displayed as red lines and orange boxes, respectively). This figure includes all measurement days with a PTR-MS (i.e. from 1 to 29 March 2015).**

[Figure]

**Figure S4: Time series of particle number $N_{PSM}$, $N_{DMPS}$ and CS in comparison with suspected parameters controlling NPF events (SO$_2$, H$_2$SO$_4$, isoprene and monoterpenes) and accumulated time series of PM$_1$ contribution.**

The color code highlights NPF event days and non-event days (grey periods). Red periods represent NPF1 event days with anthropogenic origin. Orange periods represent NPF2 event days both with mixed origins (anthropogenic and biogenic). Blue and green periods are respectively for NPF events of marine (NPF3) and biogenic origin (NPF4). Organic aerosol (OA) factors: HOA - hydrogen-like OA; SV-OOA – semi-volatile oxygen-like OA; LV-OOA – low-volatile oxygen-like OA. Time is given in local time (UTC + 2 h).

---

## Author Response (AR2)

Dear co-editor,

First, we would like to thank you for your comment. We have revised the manuscript entitled "Driving parameters of biogenic volatile organic compounds and consequences on new particle formation observed at an Eastern Mediterranean background site" according to yours comments. Kindly find below our response to the comments.

Sincerely yours,

Cécile Debevec

**In the Manuscript**

Page 4, line 30 :          "Northen" was changed by "Northern"

Page 8, line 23 :          "Planet Boundary Layer" was changed by "Planetary Boundary Layer"

Page 13, lines 15 & 23 :   "the vertical mixing" was changed by "vertical mixing"

Page 13, line 30 & Page    "significant" was changed by "significantly high"
14, line 5 :

Page 14, line 3 :          "low wind condition" was changed by "low wind conditions"

**In the Supplement**

Page 2, line 1 :           "Planet Boundary Layer" was changed by "Planetary Boundary Layer"

Page 2, line 2 :           "PBL height effect" was changed by "the PBL height effect"

Page 2, line 24 :          "that is often closer to" was changed by "that are often closer to"